# Diversity and Functional Potential of Gut Bacteria Associated with the Insect *Arsenura armida* (Lepidoptera: Saturniidae)

**DOI:** 10.3390/insects16070711

**Published:** 2025-07-10

**Authors:** María Griselda López-Hernández, Reiner Rincón-Rosales, Clara Ivette Rincón-Molina, Luis Alberto Manzano-Gómez, Adriana Gen-Jiménez, Julio Cesar Maldonado-Gómez, Francisco Alexander Rincón-Molina

**Affiliations:** 1Laboratorio de Ecología Genómica y Agricultura Regenerativa, Tecnológico Nacional de México, Instituto Tecnológico de Tuxtla Gutiérrez, Tuxtla Gutiérrez 29050, Chiapas, Mexico; griseldalohez@gmail.com (M.G.L.-H.); reiner.rr@tuxtla.tecnm.mx (R.R.-R.); clara.rm@tuxtla.tecnm.mx (C.I.R.-M.); contacto@3rbiotec.com (L.A.M.-G.); d10270415@tuxtla.tecnm.mx (A.G.-J.); d07270254@tuxtla.tecnm.mx (J.C.M.-G.); 2Departamento de Innovación y Desarrollo, 3R Biotec SA de CV, Tuxtla Gutiérrez 29000, Chiapas, Mexico

**Keywords:** gut microbiota, symbiotic bacteria, *Arsenura armida*, cellulose degradation

## Abstract

Insects rely on the microorganisms that inhabit their gut to break down food and absorb nutrients. These microorganisms play a crucial role in converting complex dietary substances into simpler compounds essential for the insect’s growth and development. This study examines the bacterial communities in the gut of *Arsenura armida* (Lepidoptera: Saturniidae) larvae, an insect native to southeastern Mexico and consumed by local communities. Over 120 bacterial strains were isolated from various gut sections, with most being rod-shaped and involved in producing substances like sugars and enzymes. Some strains showed the ability to degrade cellulose, a complex plant component. Members of the Bacillota phylum were particularly abundant, supporting the larva digestion and demonstrating potential for industrial applications. This research highlights the vital role of gut bacteria in insect survival and their potential use in biotechnological fields such as food production, recycling, and plant material processing.

## 1. Introduction

Insects are essential biological resources due to their ecological, cultural, and economic roles worldwide. In Mexico, diverse climates, topographies, and soils support a high diversity of edible insect species. These organisms act as primary consumers, predators, and providers of valuable products such as silk, honey, and wax. Beyond their ecological functions, insects are culturally significant in rural communities, where they have been consumed for centuries for medicinal, religious, and gastronomic purposes [1,2,3]. In Chiapas, southeastern Mexico, 159 edible insect species have been recorded, including 19 Lepidoptera, whose larvae and pupae are traditionally consumed by indigenous groups [4]. Among them, *Arsenura armida* larvae—known locally as “Cuetlas” or “Zats”—are harvested in summer from *Heliocarpus appendiculatus* and are highly valued for their cultural and culinary relevance [5].

*Arsenura armida*, a member of the Lepidoptera order and Saturniidae family, was first described by Pieter Cramer in 1779. Its larval stage lasts 40–60 days, with individuals reaching 12–14 cm in length and displaying green, black, and yellow ring patterns that provide camouflage against predators [4,6,7,8]. Morphologically, larvae exhibit a segmented body with a head bearing sensory organs and mandibles, a thorax with three pairs of true legs, and an abdomen with four pairs of prolegs [9]. The digestive system comprises three regions: the foregut (FG), midgut (MG), and hindgut (HG). In some Lepidoptera, including *Bombyx mori*, the MG functions as a major digestive site due to its alkaline environment, facilitating nutrient absorption and the enzymatic breakdown of lignocellulosic material [6,10].

Ecologically, *A. armida* larvae are herbivorous; they feed on the leaves of host trees such as *Heliocarpus donnellsmithii*, *H. appendiculatus*, and *Guazuma ulmifolia*, and typically occur in clusters of 50–100 individuals per tree [9,11]. Their feeding activity influences plant biomass and nutrient cycling, though large aggregations can cause significant defoliation. Upon reaching maturity, larvae pupate underground for 6–8 weeks, emerging as moths that contribute to pollination and ecosystem functioning [12]. Traditionally consumed for their cultural significance rather than nutritional value, these larvae embody the deep culinary heritage of southeastern Mexico [13]. As *A. armida* undergoes complete metamorphosis, the final larval instars—marked by rapid growth and intense metabolic activity—are most relevant for studying gut microbial diversity and enzymatic function. This study therefore focused on last-instar larvae as the peak stage of herbivorous activity and host–microbe interaction.

The gut microbiota of many insects is important for their survival, growth, and ecological adaptability. These microbial communities, composed of bacteria, fungi, archaea, and protozoa, fulfill critical functions in digestion, immunity, and nutrient assimilation [14]. Similar developmental traits and microbial associations have also been observed in Lepidopteran larvae, such as *Spodoptera litura* (Lepidoptera: Noctuidae), a well-studied model for host–microbiome interactions. Specifically, in some Lepidopteran species, gut bacteria can assist in breaking down complex lignocellulosic materials into simpler forms, providing the insect with essential nutrients for its development. Enzymes such as cellulases, amylases, and proteases, produced by symbiotic bacteria, facilitate the degradation of plant-derived compounds, enabling certain Lepidoptera to thrive in diverse ecological niches [15,16].

Although variability exists across taxa, the MG is often described as a primary digestive site in several Lepidopteran species, such as *Bombyx mori* and *Helicoverpa armigera*, housing microorganisms that may contribute to enzymatic digestion and the detoxification of plant metabolites [17,18,19]. These symbionts may enhance digestive efficiency and provide antimicrobial compounds that bolster immune defenses. For example, *Bacillus*, *Enterobacter*, and *Pseudomonas* have been identified as contributors to cellulose degradation and nutrient metabolism in Lepidopteran insects such as *Helicoverpa armigera* and *Bombyx mori* [18,19].

The role of gut bacteria may extend beyond digestion. Symbionts in some species participate in nitrogen fixation, vitamin synthesis, and the detoxification of plant secondary metabolites [15]. In Lepidopteran larvae, bacterial communities can recycle nitrogen, providing essential amino acids and other nutrients that are often absent from their plant-based diets [20,21]. The gut microbiota of *Dendrolimus superans* and *Lymantria dispar* contributes to essential physiological processes in their hosts, including vitamin synthesis, as evidenced by microbial shifts under plant alkaloid stress [22].

Modern molecular techniques, such as 16S rRNA gene sequencing and metagenomics, have revolutionized the study of insect gut microbiota. High-throughput sequencing has shown that in some Lepidopteran larvae, bacterial communities are dominated by Pseudomonadota, Bacillota, Actinobacteria, and Bacteroidetes. These taxa contribute to host nutrition, enhance immune responses, and protect against environmental toxins, demon- strating their critical role in insect physiology and ecology [15,23]. The gut microbiota of *A. armida*, however, remains unexplored. Investigating its microbial diversity and enzymatic functions could reveal novel mechanisms of nutrient metabolism with potential biotechnological applications. For example, isolating cellulolytic bacteria from *A. armida* could aid in developing enzyme systems for biofuel production and agricultural waste management.

Insights into the gut microbiota of *A. armida* at the larval stage reveal a system characterized by taxonomic diversity, functional specialization, and notable enzymatic capabilities. The focus on the larval stage of *A. armida* is essential, as it represents the primary herbivorous and feeding phase of the organism’s life cycle. During this stage, the larvae consume large amounts of lignocellulosic plant material, relying heavily on their gut microbiota for digestion and nutrient assimilation [15]. The findings contribute to a broader understanding of insect-associated microbiomes, particularly in herbivorous larvae, and underscore the potential of gut bacteria for applications in biotechnology, sustainable resource management, and industrial enzyme production [24,25].

The primary objective of this study was to investigate the gut microbiota of *Arsenura armida* larvae, with a focus on the diversity and functional potential of their symbiotic bacteria. Given that the larval stage represents the peak of herbivorous activity and microbiota-mediated interactions, we aimed to characterize bacterial diversity, assess the cellulolytic capacity, and evaluate the nutrient assimilation potential using a combination of culture-dependent techniques and DNA fingerprinting methods.

## 2. Materials and Methods

### 2.1. Larvae Collection of A. armida

The larvae and adults of *A. armida* were collected from a site known as “Los Angeles” located in the municipality of Bosque, Chiapas, Mexico (17°03′42′′ N, 92°43’16′′ W; 1227 m.a.s.l.) (Figure 1). This area is characterized by a temperate climate with frequent summer rainfall, an average annual precipitation of 1220 mm, and an average annual temperature of 18 °C. The soil is of the vertisol type, shallow, and slightly acidic (pH 6.2–6.6). The vegetation in this region is classified as a semi-evergreen tropical forest, which serves as the primary habitat for *H. donnellsmithii*, *H. appendiculatus*, and *G*. *ulmifolia*. These trees are the main hosts where the larvae of *A. armida* colonize and feed.

Larvae were collected directly from the host tree *H. appendiculatus* using fine forceps, primarily during the early stages of colonization. Additionally, larvae at the ultimate instar stage were collected for comparative analysis.

The collection of larvae was carried out in 2022 during their natural occurrence season, which is limited to the months of June, July, and August, when environmental conditions and host plant availability are optimal for larval development. The taxonomic identification of *A. armida* was based on distinctive morphological characteristics, such as body size, coloration patterns, and the presence of characteristic green, black, and yellow rings, as well as their close association with specific host plants [5,6]. Identification was further supported by consultation with local entomological experts.

Live larvae were transported to the laboratory in sterile vials containing moistened paper to maintain humidity. Once in the laboratory, the total fresh weight, average diameter, and length of the larvae were measured.

### 2.2. Isolation of Bacteria from the Gut of A. armida

Twenty last-instar larvae were randomly selected and cleaned with paper towels to remove organic debris. The larvae were anesthetized via cold treatment on ice for 10 min and then sacrificed by brief freezing at −20 °C. The larvae were then surface-sterilized using 100% ethanol, followed by 70% ethanol and sterile water, each applied twice for 1 min, to eliminate external wax and dust. The larvae were sectioned into three parts: the head (foregut, FG), abdomen (midgut, MG), and anal segment (hindgut, HG). Samples of the corresponding digestive tract were obtained from each of these sections for endobacteria isolation (Figure 2).

A total of 10 g of each gut section from *A. armida* larvae was placed in 1 mL of phosphate-buffered saline (PBS). The tissue was immediately homogenized in a sterile mortar and inoculated onto plates containing Nutrient Agar (NA: 5 g of peptone, 3 g of beef extract, and 15 g of agar per liter), Tryptic Soy Agar (TSA: 15 g of tryptone, 5 g of soy peptone, 5 g of sodium chloride, and 15 g of agar per liter), and Brain Heart Infusion Agar (BHIA: 200 mg of calf brain infusion, 250 mg of beef heart infusion, 10 g of proteose peptone, 2 g of dextrose, 5 g of sodium chloride, 2.5 g of disodium phosphate, and 15 g of agar per liter). Serial dilutions ranging from 10^−1^ to 10^−8^ were prepared according to the method described by Gomes et al. [26]. A volume of 100 µL from each dilution was plated onto the culture media. Petri dishes were inoculated using sterile glass beads and incubated at 28 °C under aerobic conditions for two days. This prolonged incubation period allowed for the recovery of slow-growing environmental and gut-associated bacteria. Aerobic conditions were selected to isolate culturable bacteria with potential biotechnological applications, particularly those involved in cellulose degradation. Anaerobic and facultative anaerobic microbial populations, which are often unculturable or require specialized conditions, were explored through metagenomic analysis to capture a broader representation of the gut microbiota. Emerging bacterial colonies were subcultured onto the respective media and subsequently stored in a liquid medium supplemented with 30% glycerol at −70 °C, establishing a master bacterial bank.

### 2.3. Culture-Independent Characterization of GUT Bacteria from A. armida

A metagenomic analysis was conducted using DNA extracted from the entire gut (non-sectioned) to capture the overall bacterial community structure and functional potential. DNA was then extracted using the ZymoBIOMICS^®^ DNA Miniprep Kit (Zymo Research Corporation, Tustin, CA, USA). The DNA concentration and purity were measured using a NanoDrop^®^ spectrophotometer (ThermoFisher Scientific, Waltham, MA, USA). Amplification of the V3–V4 region of the bacterial 16S *r*RNA gene was performed with primers Bakt_341F (CCTACGGGNGGCWGCAG) and Bakt_805R (GACTACHVGGGTATCTAATCC) on an Illumina^®^ MiSeq platform with 2 × 300 paired-end reads [27]. Sequencing was conducted via Macrogen DNA sequencing services (Seoul, Republic of Korea). Bioinformatics analyses were performed using the DADA2 pipeline (v1.22.0) [28] and the Phyloseq package (v1.38.0) [29] in the open-source software Rstudio (v4.1.2) [30]. Taxonomy was assigned to the amplicon sequence variants (ASVs) using the native Bayesian classifier method with the Silva nr99 v.138 database [31]. The bacterial community structure at the phylum, order, genus, and species levels were visualized through a heat map created with the ggplot2 package (v3.3.5) [32]. The raw sequencing data have been deposited in the Sequence Read Archive (SRA) at NCBI under BioProject accession number PRJNA1218276.

### 2.4. Statistical Analysis

A Kruskal–Wallis rank-sum test followed by Dunn’s test, corrected with the Benjamini–Hochberg method, were conducted to assess statistically significant changes in the bacterial community structure at the phylum level using the software RStudio (v4.1.2) [30]. Alpha diversity indices, including Shannon, NP-Shannon, Simpson, ACE, Chao1, and Jackknife, were calculated [33].

### 2.5. Functional Predictions of the Bacterial Community

To predict the functionality of the bacterial community from *A. armida* gut amplicon metagenomic data, the PICRUSt algorithm [34] was employed. ASV data were normalized using 16S rRNA gene copy numbers. To link the ASV table and predict metagenomic functions based on the KEGG database, the R package Tax4Fun (v. 0.3.1) [35] was utilized. The predicted functionality of the bacterial community was analyzed at the levels of metabolic pathways and orthologous genes, with graphical representations generated as bar plots using the ggplot2 package.

### 2.6. Culture-Dependent Characterization of Bacteria Isolated from A. armida

The molecular characterization of bacterial isolates commenced with DNA extraction using the Roche^®^ DNA Isolation Kit for Cells and Tissues (Roche Diagnostics Corporation, Indianapolis, IN, USA), followed by quality assessment via 1% agarose gel electrophoresis. The extracted DNA was stored at −20 °C. Genomic fingerprinting was performed using the Enterobacterial Repetitive Intergenic Consensus (ERIC) technique to classify isolates based on electrophoretic banding patterns, as described by de Bruijn [36]. The primers ERIC 1R (5′-ATG TAA GCT CCT GGG GAT TCA C-3′) and ERIC 2 (5′-AAG TAA GTG ACT GGG GTG AGC G-3′) were used in an Applied Biosystems^®^ 2720 thermal cycler (Applied Biosystems, Foster City, CA, USA).

To identify the isolated bacterial strains, the 16S rRNA gene was amplified using the universal primers 27F (5′-AGAGTTTGATCMTGGCTCAG-3′) and 1492R (5′-GGTTACCTTGTTACGACTT-3′), generating fragments of approximately 1500 bp in length. Amplified products were digested with the restriction enzyme *Rsa*I (5′-GT^AC-3′) (Fast Digest, Thermo Scientific^®^) for Amplified Ribosomal DNA Restriction Analysis (ARDRA) and visualized on 3% agarose gel stained with ethidium bromide. Purified PCR products were quantified using a NanoDrop 2000c spectrophotometer (Thermo Scientific^®^) and submitted for Sanger sequencing. Richness (*d*) and diversity (*H)* indices were calculated based on BOX and ARDRA genetic profiles. The obtained sequences were aligned and edited using BIOEDIT v.7.2.5 and analyzed through BLAST v.2.2.25 for taxonomic identification in the NCBI database (https://blast.ncbi.nlm.nih.gov/Blast.cgi, accessed on 10 March 2025). Phylogenetic trees were constructed using MEGA v.10.0, applying the Neighbor-Joining method with the Tamura-Nei model [37]. All sequences isolated from *A. armida* were deposited in GenBank under accession numbers KX389672 to KX389699 and KX827623 to KX827626. The sequences were deposited in two different batches due to the timing of data acquisition and processing. Additionally, the cellular morphology of the bacteria and the colony morphology formed on culture media were studied using light microscopy and Gram differential staining for a representative strain of each of the identified species.

### 2.7. Cellulolytic Enzyme Production Capacity

The ability to produce cellulolytic enzymes (cellulase) was evaluated using yeast mannitol agar (YMA) medium supplemented with 1% (*w*/*v*) carboxymethylcellulose (CMC). Plates were inoculated with 10 μL of a bacterial suspension at a concentration of 1 × 10^6^ CFU/mL using bacterial strains isolated from the gut of *A. armida* larvae, and incubated for three days at 28 °C. After incubation, bacterial colonies were removed by washing with distilled water, and the plates were stained with 0.1% Congo Red solution for 30 min. Subsequently, plates were washed with 1M NaCl until clear lysis halos, indicative of cellulase production, became visible [38].

### 2.8. Cellulase Activity Assay

Carboxymethylcellulase (CMC) activity was determined by measuring the release of reducing sugars from CMC hydrolysis. A standard glucose calibration curve was prepared under assay conditions using 1% (*w*/*v*) glucose. A minimal CMC medium was prepared, consisting of 0.5 g/L K_2_HPO_4_, 0.25 g/L MgSO_4_, 2 g/L carboxymethylcellulose, and 3 g/L NH_4_Cl. Each flask was inoculated with the respective bacterial strains isolated from the gut of *A. armida* larvae and incubated at 27 °C with agitation at 120 rpm for 48 h.

At the end of the incubation period, the reducing sugar content in the culture supernatant was determined. One milliliter of the supernatant was mixed with 1.5 mL of dinitrosalicylic acid (DNS) reagent, heated until boiling for 10 min, and then cooled to room temperature. The reaction mixture was adjusted to 4 mL with distilled water, and optical density (OD) was measured at 535 nm using a spectrophotometer [39,40].

## 3. Results

### 3.1. Morphometric Characteristics of A. armida Larvae

Fifty last-instar larvae of *A. armida* were analyzed to determine their total length, diameter, weight, and visual and physical characteristics. The larvae exhibited smooth, cylindrical bodies with vibrant green coloration accented by striking black and yellow rings along their segments. Morphometric analysis revealed an average total length of 11.75 ± 0.35 cm, a mean diameter of 1.36 ± 0.19 cm, and an average weight of 11.55 ± 0.53 g.

Additionally, the larvae displayed a healthy, robust appearance, with a firm yet elastic cuticle. The combination of morphometric data and visual traits confirmed that the larvae were at the stage of maximum growth and development.

### 3.2. Bacterial Community Structure Associated with the Gut of A. armida

The bacterial community associated with the gut of *A. armida* larvae displayed a dominant composition across taxonomic levels (Figure 3). The obtained sequences had an average length of 300 bp, ensuring sufficient resolution for taxonomic classification and functional analysis. At the phylum level, Bacillota (Firmicutes) dominated the community (*p* < 0.05), accounting for 96% of the total abundance, followed by Proteobacteria (Pseudomonadota) and Actinobacteria, each contributing 2%. Although the overall differences in phylum composition were statistically significant according to the Kruskal–Wallis test (*p* = 0.0117), pairwise comparisons using Dunn’s test with Benjamini–Hochberg correction revealed no statistically significant differences between individual phyla (adjusted *p* > 0.05). At the order level, Clostridiales represented 94% of the bacterial population, with Lactobacillales and Rhizobiales contributing 2% and 1%, respectively. At the genus level, an unclassified genus within the Lachnospiraceae family was the most prevalent, representing 94% of the community, while *Methylobacterium* and *Enterococcus* accounted for 1% and 2%, respectively. A considerable portion of the community remained unclassified at the species level (94.36%), underscoring the high microbial diversity within the larval gut ecosystem. However, a few species were identified, including the *Enterococcus faecium* group, *Quadrisphaera granulorum*, *Methylobacterium komagatae*, *Methylobacterium aerolatum*, *Methylobacterium oryzae* group, *Methylobacterium extorquens* group, *Methylobacterium marchantiae* group, *Aureimonas jatrophae* group, *Kineococcus aurantiacus* group, *Pseudoglutamicibacter cumminsii*, *Corynebacterium stationis*, and *Leucobacter tardus*, among others. Additionally, the observed low Shannon diversity index values combined with high Simpson indices suggest low evenness within the microbial community, indicating dominance by one or a few bacterial taxa, such as Bacillota (Clostridiales and Lachnospiraceae_uc), which disproportionately contribute to the community structure.

### 3.3. Alpha Diversity of the GUT Microbiota Associated with A. armida Larvae

The alpha diversity analysis of the gut bacterial community in *A. armida* larvae, assessed through the Shannon, NP-Shannon, Simpson, ACE, Chao1, and Jackknife indices, revealed variations in species richness and diversity across biological replicates (Zats_1, Zats_2, and Zats_3). The richness estimators ACE (163 ± 43), Chao1 (127 ± 6), and Jackknife (135 ± 6) indicated an abundant bacterial community, with Zats_3 exhibiting the highest estimated richness, suggesting a broader microbial reservoir, while Zats_2 displayed the lowest. The Shannon (0.25 ± 0.08) and NP-Shannon (0.26 ± 0.08) indices, which account for species evenness and diversity, showed moderate variation, with Zats_1 presenting the highest diversity, followed by Zats_3 and Zats_2. The Simpson (0.93 ± 0.03) index, which reflects community dominance, suggested that a few taxa were overwhelmingly abundant, reinforcing a structured and specialized microbiota.

### 3.4. Functional Metabolic Potential of the Gut Microbiota in A. armida Larvae

Functional metagenomic analysis of the gut microbiota associated with *A. armida* larvae revealed a wide array of metabolic and biochemical functions. KEGG pathway analysis (Figure 4a) indicated the presence of pathways involved in carbohydrate metabolism, amino acid biosynthesis, secondary metabolite production, and energy generation.

The microbial pathways related to methane metabolism, metabolic adaptation to diverse environments, and oxidative phosphorylation were among the most highly enriched, reflecting the functional diversity of the gut microbiota.

Genes associated with glycolysis/gluconeogenesis, the pentose phosphate pathway, and pyruvate metabolism were identified. Genes involved in the biosynthesis of essential amino acids, including lysine and phenylalanine, were also detected.

Additional pathways identified included peptidoglycan biosynthesis, pentose and glucuronate interconversions, and nucleotide metabolism. Genes related to flagellar assembly and quorum sensing were present as well.

The analysis of orthologous genes (Figure 4b) revealed the presence of transport systems, multidrug resistance mechanisms, and stress response genes. Two-component regulatory systems, transcriptional regulators, and DNA recombinases were also identified. Additionally, genes encoding ABC transporters, multiple sugar transporters, and chemotaxis proteins were detected. Biosynthetic pathways for secondary metabolites and antibiotics were present as well.

### 3.5. Diversity and Genetic Identification of Bacterial Isolates from the Gut of A. armida

The diversity and abundance of bacterial isolates from the gut of *A. armida* larvae were assessed using the Shannon–Weaver diversity index (Table 1). A total of 96 isolates were classified into 18 distinct groups based on ARDRA profiles. The midgut (MG) exhibited the highest bacterial richness (*d* = 5.57) and diversity (H = 2.48), with 43 isolates representing 44.45% of the total relative abundance, suggesting a complex and metabolically active microbial community. The foregut (FG) displayed moderate diversity, comprising 37 isolates grouped into seven ARDRA profiles, with a richness index of 4.78 and a diversity index of 2.12, accounting for 38.89% of the total abundance. In contrast, the hindgut (HG) harbored the lowest microbial diversity, with only 16 isolates distributed among three ARDRA profiles, showing the lowest richness (*d* = 1.59) and diversity index (*H* = 0.70), contributing 16.67% to the total bacterial composition.

The phylogenetic analysis of 16S *r*RNA gene sequences from bacterial strains isolated from the gut of *A. armida* larvae revealed a taxonomically diverse microbiota, predominantly classified within the phyla Pseudomonadota and Bacillota (Table 2). A total of 18 bacterial species were successfully identified, each showing high sequence similarity (≥88.7%) to previously described species in the NCBI database.

The phylum Pseudomonadota accounted for most of the isolates (72%), with *Serratia marcescens* emerging as the most frequently detected species across gut sections, exhibiting sequence identities between 96.8% and 99.6%. This species was particularly prevalent in the MG and HG. Additional representatives of Pseudomonadota included *Enterobacter cloacae* (99.0%) and *Pseudomonas putida* (99.0%).

The phylum Bacillota comprised 28% of the identified taxa, with isolates predominantly affiliated with *Bacillus* species, including *Bacillus subterraneus* (97.6%) and *Bacillus odysseyi* (88.7%). These strains were primarily recovered from the FG and MG. Additionally, *Enterococcus casseliflavus* (98.4%) was detected.

The gut microbiota of *A. armida* exhibited spatial compartmentalization, with the FG and MG showing the highest taxonomic diversity. In contrast, the HG displayed reduced diversity and was predominantly composed of *Serratia* and *Enterobacter* species (Figure 5).

Likewise, the phylogenetic tree shows that several Serratia strains isolated from the gut of *Arsenura armida* (e.g., AAM18, AAP15, AAT06, and AAM09) cluster closely with *Serratia marcescens* reference strains from GenBank, such as PSB-36, DSM 30121, and RK26, with high bootstrap support (Appendix A). This indicates high genetic similarity based on 16S rRNA gene sequences. However, some isolates (e.g., AAM11, AAM12, and AAM08) form well-supported separate branches, suggesting intraspecific variability or potential novel *Serratia* lineages associated with this Lepidopteran host. Nonetheless, further research, including whole-genome sequencing and additional phylogenetic markers, is necessary to confirm the presence of new species.

In the case of *Bacillus* isolates, the phylogenetic tree based on 16S *r*RNA gene sequences shows that several strains from the *Arsenura armida* gut, such as AAM13, AAT05, and AAT04, form distinct, well-supported clades (Appendix A). For example, AAM13 clusters closely with *Bacillus thioparans* (71% bootstrap), while AAT05 appears closely related to *B. subterraneus* and *B. selenatarsenatis* based on their phylogenetic placement. Meanwhile, AAT04 forms a separate clade near *Lysinibacillus halotolerans* (75% bootstrap), suggesting notable phylogenetic diversity and the presence of well-characterized *Bacillus* species in the larval gut microbiota.

### 3.6. Cellulase Activity of Bacterial Strains Isolated from the Gut of A. armida

The cellulolytic activity of bacterial strains isolated from different gut sections of *Arsenura armida* larvae was evaluated using both qualitative (CMC plate assay) and quantitative (reducing sugar production) methods. A total of 18 bacterial strains were tested for their ability to hydrolyze carboxymethylcellulose (CMC) and produce reducing sugars (mg/mL) (Table 3). Of these, thirteen strains exhibited positive cellulase activity, while five strains showed no detectable activity (ND: not detected).

The CMC plate assay revealed that midgut-derived strains demonstrated the strongest extracellular cellulase activity, with *Enterobacter* sp. AAM10 displaying the largest clear zone diameter (16.2 ± 0.81 mm), followed by *Serratia marcescens* AAM08 (14.5 ± 0.91 mm) and *Bacillus* sp. AAM13 (14.0 ± 0.88 mm). In comparison, foregut isolates such as *Enterobacter cloacae* AAT03 (13.1 ± 0.81 mm) and *Pseudomonas putida* AAT01 (12.3 ± 0.56 mm) showed moderate halo formation, while hindgut strains had lower diameters, with *S. marcescens* AAP15 reaching only 11.8 ± 0.43 mm. Several strains showed no halo formation, indicating an absence of extracellular cellulase activity. These data suggest that the midgut is the primary site of microbial cellulose degradation in *A. armida* larvae. Consistent with the qualitative results, the highest reducing sugar concentrations were recorded for *Enterobacter* sp. AAM10 (35.185 mg/mL), *Bacillus* sp. AAM13 (29.540 mg/mL), and *S. marcescens* AAM08 (28.599 mg/mL), all isolated from the midgut. Foregut isolates such as *E. cloacae* AAT03 (26.031 mg/mL) and *P. putida* AAT01 (23.895 mg/mL) also exhibited considerable enzymatic activity. Notably, Enterobacter sp. AAM10 produced approximately 2.5 times more reducing sugars than *S. marcescens* AAT06 (13.934 mg/mL), underscoring its strong cellulolytic capability.

Conversely, five strains—*Enterococcus* sp. AAT02, *Bacillus* sp. AAT04, *Bacillus* sp. AAT05, *Enterobacter asburiae* AAP16, and *Acinetobacter quilouiae* AAP17—exhibited no detectable cellulase activity, suggesting that not all members of the gut microbiota are involved in cellulose degradation. Among the hindgut isolates, only *S. marcescens* AAP15 showed moderate enzymatic potential, producing 24.172 mg/mL of reducing sugars.

The bar plot reveals distinct regional patterns in cellulolytic activity among gut-derived bacterial strains of *A. armida* larvae (Figure 6). Midgut isolates exhibited the highest reducing sugar production, with *Enterobacter* sp. AAM10 (35.185 ± 0.33 mg/mL), *Bacillus* sp. AAM13 (29.540 ± 0.25 mg/mL), and *Serratia marcescens* AAM08 (28.599 ± 0.36 mg/mL) showing the most pronounced activity. In contrast, foregut strains like *E. cloacae* AAT03 and *P. putida* AAT01 produced moderate levels (26.031 ± 0.47 and 23.895 ± 0.36 mg/mL, respectively), while hindgut strains generally showed lower activity, with only *S. marcescens* AAP15 reaching 24.172 ± 0.65 mg/mL. These results underscore the midgut as the primary site for microbial-mediated cellulose degradation, driven by highly active strains with significant bioconversion potential.

## 4. Discussion

This study provides a comprehensive characterization of the bacterial communities associated with *A. armida* larvae, integrating taxonomic diversity, functional metabolic potential, and cellulolytic activity to elucidate the role of gut bacteria in host digestion. The morphometric analysis of *A. armida* larvae confirmed their developmental maturity, showing consistent size and weight characteristics. The observed morphological traits, such as a firm and elastic cuticle, indicate that the larvae were in a physiologically healthy condition with no apparent signs of bacterial infection, suggesting that the isolated gut bacteria are likely commensal or non-pathogenic. Similar developmental traits and reliance on microbial assistance have been reported in other Lepidopteran larvae, such as *Spodoptera litura* (Lepidoptera: Noctuidae), where the bacterial diversity varies across different developmental stages, correlating with microbial colonization capacity [41,42,43]. These structural features have been linked to the capacity of the gut to host dense and metabolically active microbial communities, especially in species such as *Brithys crini* and *Spodoptera litura*, which feed on recalcitrant plant tissues and face significant enzymatic challenges [42].

The bacterial community composition was predominantly structured by Bacillota at the phylum level (96%), followed by Proteobacteria (Pseudomonadota) and Actinobacteria (2% each). At the order level, Clostridiales (94%) was the most prevalent, while at the genus level, Lachnospiraceae_uc (94%) was the most abundant taxon. This taxonomic profile is consistent with previous findings in the Lepidopteran gut microbiota, where Bacillota dominate due to their role in cellulose degradation and fermentation [15,42]. The diversity and abundance of bacterial isolates from the gut of *A. armida* revealed that microbial richness varied significantly across gut sections, with the midgut (MG) exhibiting the highest diversity (H = 2.48), followed by the foregut (FG) (H = 2.12) and hindgut (HG) (H = 0.70). The observed compartmentalization along the digestive tract of *Arsenura armida* may indicate functional differentiation among gut regions. Our results suggest that the midgut harbors a greater diversity of culturable microbes and may represent the principal site for enzymatic digestion and microbial interaction. While previous studies in other insect taxa have identified the hindgut as a specialized site for fermentation processes [44,45], our morphological observations and culture-based data do not support this configuration in *A. armida*. Consequently, any assumption regarding fermentative activity in the hindgut of this species should be approached with caution and considered hypothetical until further functional or metagenomic evidence is available.

This metabolic similarity may be considered a case of convergent functional adaptation, as both *A. armida* larvae and termites depend on lignocellulose-rich diets that likely exert similar selective pressures on their gut microbiota, favoring microbial communities capable of efficiently processing plant-derived polysaccharides. Comparable gut regionalization has been described in xylophagous beetles such as *Odontotaenius disjunctus*, described by Illiger in 1800 and belonging to the family Passalidae within the order Coleoptera, in which specific gut compartments harbor distinct bacterial communities linked to lignocellulose processing [46,47]. Such parallels suggest a conserved digestive strategy among herbivorous insects feeding on fibrous plant material. The observed dominance of Bacillota, particularly *Clostridiales*, suggests that the gut of *A. armida* supports a fermentative environment, especially in posterior regions [48]. A similar pattern has been reported in *Reticulitermes flavipes*, a termite species described by Kollar in 1837 and classified within the order Isoptera and family Rhinotermitidae, where microbial fermentation significantly contributes to volatile fatty acid production and host energy supply [48]. This evidence supports the ecological convergence of the gut microbiota composition in insects with similar dietary constraints [48]. The divergence between bacterial taxa identified through culture-dependent methods (e.g., Pseudomonadota such as Enterobacter, Pseudomonas, and Serratia) and those detected via metagenomic analysis (predominantly Bacillota) stems from the intrinsic limitations of cultivation-based approaches. These methods tend to favor fast-growing, aerotolerant microorganisms under nutrient-rich, aerobic laboratory conditions. In contrast, metagenomic techniques provide an unbiased overview of the entire microbial community, including non-culturable and obligate anaerobic taxa that are often dominant in insect gut environments. Thus, the integration of both strategies yields a more comprehensive perspective, combining the functional characterization of culturable isolates with a broader ecological understanding of the gut microbiota composition.

Metagenomic analysis revealed that the gut microbiota of *A. armida* harbors key metabolic pathways involved in carbohydrate metabolism, amino acid biosynthesis, and microbial adaptation. The enrichment of glycolysis, gluconeogenesis, and pentose phosphate pathways indicates an optimized system for energy production and carbon utilization, similar to findings in termite gut microbiomes [49,50]. Additionally, the identification of genes related to peptidoglycan biosynthesis, quorum sensing, and flagellar assembly suggests that gut bacteria employ structural and communication mechanisms to maintain stability within the host environment [51,52].

The dynamic and functionally specialized nature of the *A. armida* gut microbiota supports the host’s ability to digest plant-derived substrates efficiently. Similar metabolic features have been identified in the guts of caterpillars such as *Ostrinia nubilalis*, a species described by Hübner in 1976 (Crambidae) within the order Lepidoptera, where microbial genes related to carbohydrate-active enzymes (CAZymes) play a crucial role in supplementing the host’s enzymatic digestion of plant cell walls [45]. These comparisons reinforce the idea that functional convergence among gut microbiomes occurs in insects feeding on structurally complex diets. Notably, the detection of pathways for amino acid biosynthesis suggests a potential microbial contribution to host nitrogen metabolism, as reported in symbiotic systems such as the gut of *Blattella germanica*, a cockroach species described by Linnaeus in 1767 and classified within the family Ectobiidae of the order Blattodea, where the microbial synthesis of essential amino acids complements dietary deficiencies [53]. Furthermore, the presence of quorum sensing and flagellar genes may indicate active microbial colonization dynamics and interspecies communication mechanisms.

Phylogenetic analysis of 16S rRNA sequences revealed that the bacterial strains associated with the gut of *A. armida* predominantly belong to the phyla Pseudomonadota and Bacillota. Among the identified strains, *Serratia marcescens* appears to be particularly prevalent in the midgut (MG) and hindgut (HG), regions where enzymatic activity is typically concentrated [54]. This distribution pattern suggests that *S. marcescens* may play a significant role in cellulose hydrolysis and metabolic interactions within the gut microbiota. Other metabolically versatile bacteria, such as *Enterobacter cloacae* and *Pseudomonas putida*, have also been associated with plant polymer degradation, highlighting their potential involvement in the digestion process. Additionally, the presence of *Bacillus subterraneus* and *B. odysseyi* in the foregut (FG) and midgut (MG) supports the notion that Bacillota may play a key role in cellulose degradation, emphasizing the differential contributions of bacterial consortia to digestion across gut compartments. Members of *Serratia*, *Pseudomonas*, and *Enterobacter* have also been isolated from the guts of other herbivorous insects such as locusts (*Locusta migratoria*), where they participate in lignocellulose degradation and nitrogen cycling [55]. This pattern of recurrence across insect taxa points to their potential core role in shaping the gut microbiota of herbivorous species. Given the conserved occurrence of these genera in diverse insect hosts, they may represent key taxa with flexible metabolic repertoires enabling adaptation to various plant-based diets. Their detection in multiple gut compartments also suggests niche partitioning within the digestive tract, contributing to overall digestive efficiency.

Cellulolytic activity assays confirmed that 13 out of 18 bacterial strains exhibited positive cellulase activity, with the midgut (MG) harboring the most active isolates. *Enterobacter* sp. AAM10 showed the highest reducing sugar production (35.185 mg/mL), followed by *Bacillus* sp. AAM13 (29.540 mg/mL) and *Serratia marcescens* AAM08 (28.599 mg/mL). These findings suggest that microbial-assisted fiber degradation in *A. armida* is concentrated in the MG. This contrasts with xylophagous insects such as termites and wood-feeding beetles, in which the hindgut is the primary site of fermentation due to its enlarged structure and anaerobic environment [56,57]. This observation highlights potential differences in gut physiology and microbial function between Lepidoptera and other detritivorous or wood-feeding taxa. Further research involving direct enzymatic testing of gut contents is required to confirm the in vivo compartmentalization of cellulolytic activity. *Enterobacter cloacae* AAT03 and *Pseudomonas putida* AAT01, both isolated from the foregut (FG), exhibited significant cellulolytic activity. Although the midgut and hindgut are commonly associated with cellulose degradation, the presence of active cellulolytic strains in the FG suggests that the initial stages of lignocellulose breakdown may begin earlier in the digestive tract. This supports the idea that cellulose degradation is a spatially coordinated process involving multiple gut regions, as also observed in wood-feeding insects like *P. angustipennis* and *O. disjunctus* [58]. Interestingly, five bacterial strains, including *Enterococcus* sp. AAT02 and *Acinetobacter quilouiae* AAP17, exhibited no detectable cellulase activity, reinforcing the concept that gut microbiomes comprise both cellulolytic and non-cellulolytic taxa that interact to optimize host digestion [59].

The presence of highly active cellulolytic strains from *Enterobacter, Serratia, Bacillus,* and *Pseudomonas* underscores their ecological role in host nutrition and digestion, while also highlighting their biotechnological potential. The ability of these bacteria to degrade lignocellulosic biomass efficiently suggests their potential for applications in industrial biofuel production and agricultural waste management [60,61]. The distinct microbial compartmentalization observed in the gut further supports the hypothesis that specific bacterial consortia are adapted to different digestive processes, enhancing host efficiency at utilizing plant-based diets [46]. Future research should investigate the synergistic interactions among these gut bacteria, their enzymatic pathways, and their potential applications in biomass degradation and bioethanol production. Specifically, the ability of certain strains to produce cellulases and related carbohydrate-active enzymes positions them as promising candidates for breaking down lignocellulosic material into fermentable sugars, an essential step in the bioethanol production process [45,62]. Understanding these microbial enzymatic systems could inform the development of more efficient bioconversion strategies for renewable energy.

Future studies should focus on elucidating the molecular mechanisms underlying host–microbiome interactions and the functional role of bacterial consortia in lignocellulose degradation to fully harness the biotechnological potential of *A. armida* gut bacteria.

## 5. Conclusions

This study provides a detailed characterization of the gut microbiota in *Arsenura armida* larvae, revealing a taxonomically diverse and functionally relevant microbial community. The midgut showed the greatest microbial diversity and harbored numerous actively cellulolytic bacteria likely involved in fiber degradation. However, cellulolytic microorganisms were also present in the foregut and hindgut. Functional metagenomic predictions identified pathways linked to carbohydrate metabolism and amino acid biosynthesis, suggesting microbial contributions to digestion and nutrient assimilation. Cellulolytic activity assays confirmed the roles of *Enterobacter*, *Serratia*, *Bacillus*, and *Pseudomonas*, with *Enterobacter* sp. AAM10 showing the highest enzymatic activity. By integrating culture-dependent and metagenomic approaches, this work underscores the ecological relevance and biotechnological potential of gut microbes in herbivorous Lepidoptera. Future research should focus on the functional validation of microbial enzymes and evaluate their applications in lignocellulose bioconversion and sustainable agriculture.

## Figures and Tables

**Figure 1 insects-16-00711-f001:**
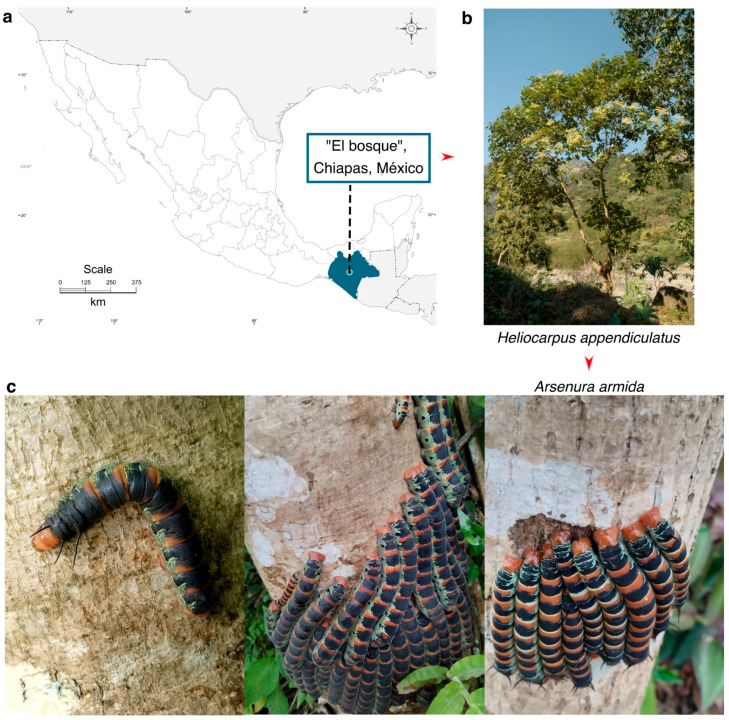
Geographic location of the collection site for *A. armida* (Zats) larvae in their natural habitat. (**a**) Map showing the collection site. (**b**) Host tree. (**c**) Final-instar larvae of *A. armida* observed feeding and aggregating on the trunk of *H. appendiculatus*, displaying their characteristic coloration and gregarious behavior during the feeding phase.

**Figure 2 insects-16-00711-f002:**
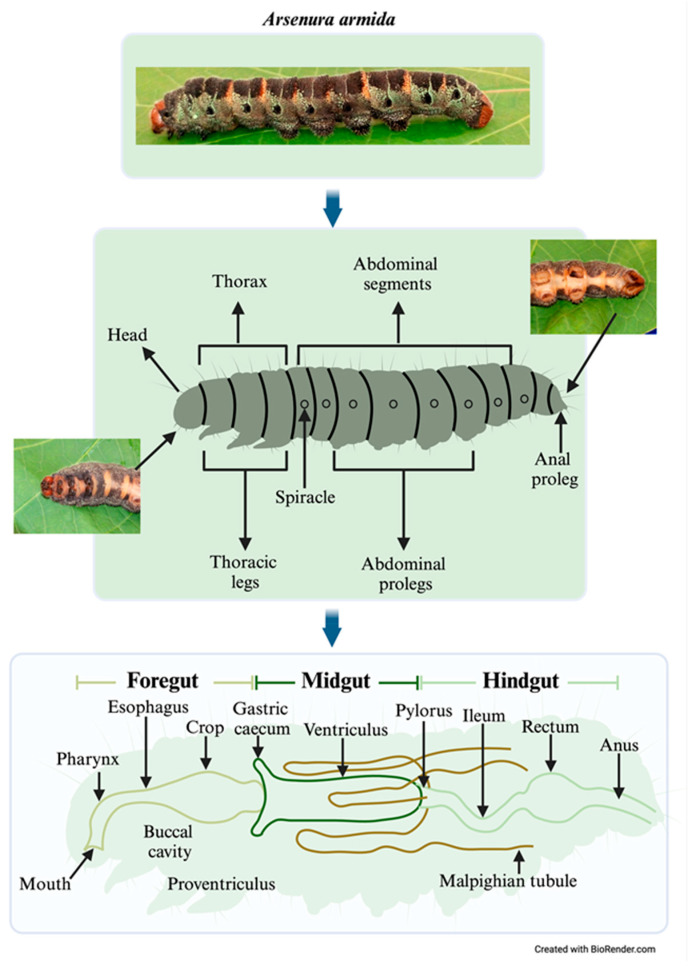
Anatomical correspondence between external body segments and internal gut regions in *Arsenura armida* larvae, highlighting digestive tract organization and sampling sites.

**Figure 3 insects-16-00711-f003:**
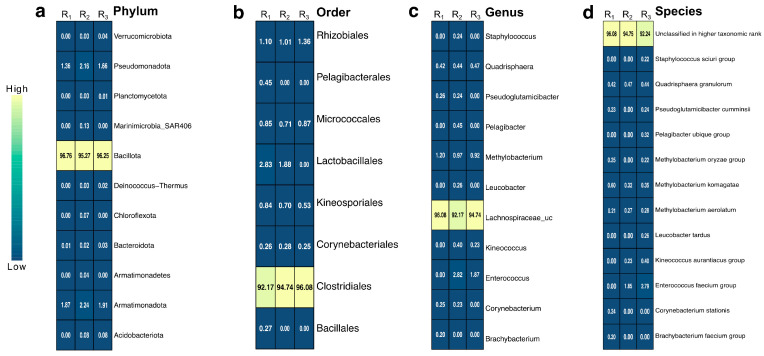
Heatmap of relative abundance (%) of bacterial taxa with values greater than 0.2%, based on 16S rRNA gene sequencing of biological triplicates. Taxonomic profiles are shown at four hierarchical levels: (**a**) Phylum, (**b**) Order, (**c**) Genus, and (**d**) Species. Columns represent replicates (R_1_, R_2_, and R_3_), and each cell shows the observed percentage of relative abundance for the corresponding taxon. Color intensity reflects abundance levels, with lighter shades indicating higher abundance.

**Figure 4 insects-16-00711-f004:**
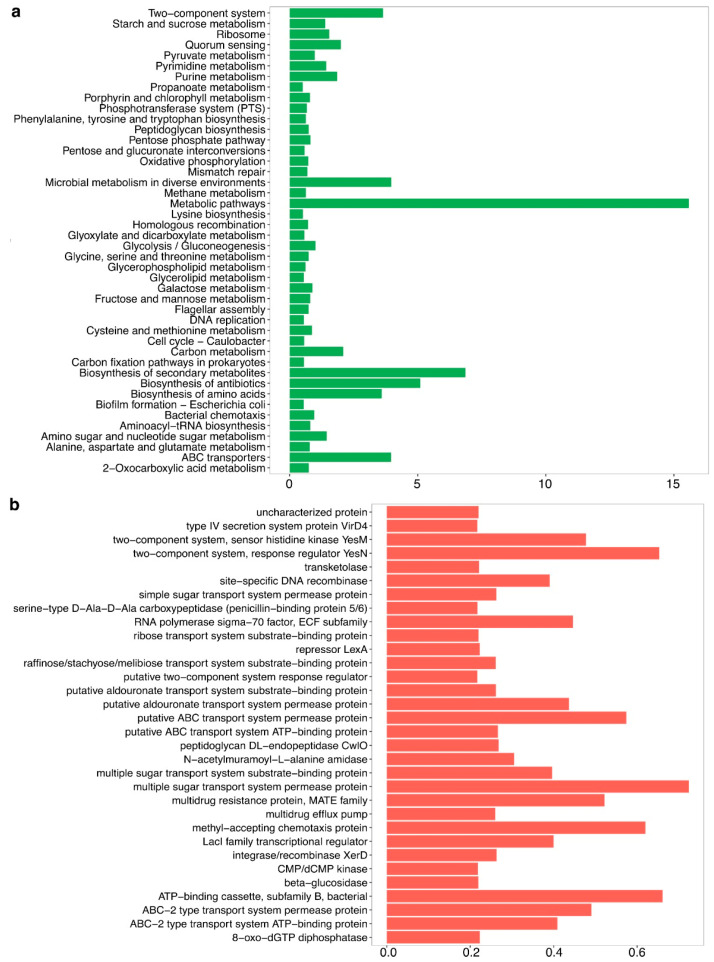
Functional metabolic potential of the gut microbiota in *A. armida* larvae. (**a**) Predicted metabolic pathways based on KEGG orthologs, highlighting functions related to carbohydrate metabolism, amino acid biosynthesis, energy production, and microbial adaptation, expressed as relative abundance. (**b**) Functional gene annotations representing predicted protein functions, including transport systems, regulatory proteins, enzymatic activities, and antibiotic resistance mechanisms, also expressed as relative abundance.

**Figure 5 insects-16-00711-f005:**
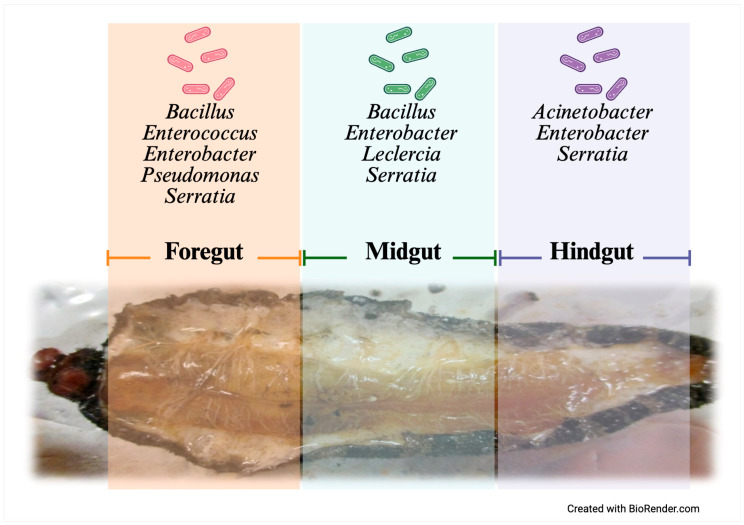
Gut compartmentalization and distribution of culturable bacteria in *A. armida* larvae.

**Figure 6 insects-16-00711-f006:**
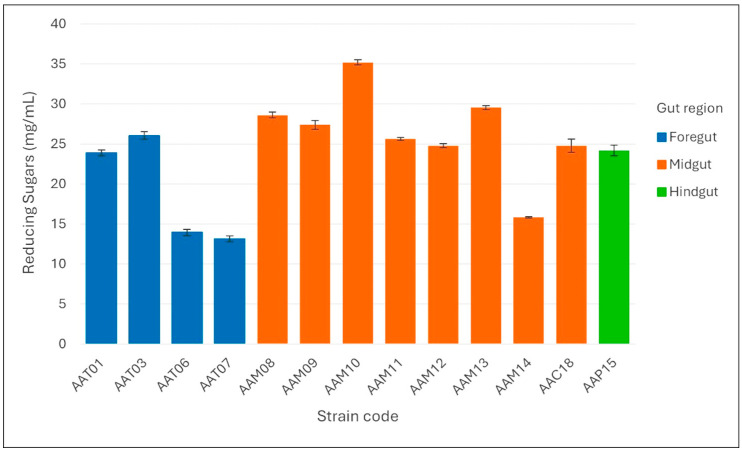
Reducing sugar production (mg/mL) by cellulolytic bacterial strains isolated from different gut regions (foregut, midgut, and hindgut) of *Arsenura armida* larvae. Bars represent the mean values ± standard deviation of triplicate measurements.

**Table 1 insects-16-00711-t001:** Diversity and abundance of bacterial species isolated from the gut of *A. armida* larvae.

Sample Source	No. of Isolates	No. of Groups ARDRA Profiles ^a^	Relative Abundance (%)	Shannon–Weaver Index ^b^	
				Richness (*d*)	Diversity (*H*)
Foregut	37	7	38.89	4.77	2.12
Midgut	43	8	44.45	5.57	2.48
Hindgut	16	3	16.67	1.59	0.70
Total	96	18	100		

^a^ ARDRA profiles, amplified *r*RNA restriction analysis obtained with *Hinf*I restriction enzyme. ^b^ Estimated using the Shannon–Weaver Index [41].

**Table 2 insects-16-00711-t002:** Phylogenetic affiliation of bacterial strains isolated from *A. armida*.

Strain	Accession Number	Closest NCBI Match/Species Identity (%)	Gut Section	Phylum
*Pseudomonas* sp. AAT01	KX389675	*Pseudomonas putida* L3 (KT767824)/99.0 ^a^	(FG)	Pseudomonadota
*Enterococcus* sp. AAT02	KX389697	*Enterococcus casseliflavus* (AF039903)/98.4	(FG)	Bacillota
*Enterobacter* sp. AAT03	KX389696	*Enterobacter cloacae* RCB970 (KT261182)/99.0	(FG)	Pseudomonadota
*Bacillus* sp. AAT04	KX389689	*Bacillus odysseyi* 34hs1 (AF526913)/88.7	(FG)	Bacillota
*Bacillus* sp. AAT05	KX389692	*Bacillus subterraneus* DSM13966^T^ (FR733689)/97.6	(FG)	Bacillota
*Serratia* sp. AAT06	KX389674	*Serratia marcescens* RK26 (KC790279)/99.0	(FG)	Pseudomonadota
*Enterobacter* sp. AAT07	KX389687	*Enterobacter ludwigii* RCB31 (KT260531)/98.0	(FG)	Pseudomonadota
*Serratia* sp. AAM08	KX389672	*Serratia marcescens* KRED (AB061685)/99.5	(MG)	Pseudomonadota
*Serratia* sp. AAM09	KX389698	*Serratia marcescens* DSM 30121 (AJ233431)/97.8	(MG)	Pseudomonadota
*Enterobacter* sp. AAM10	KX389686	*Enterobacter asburiae* R23 (KM019904)/93.0	(MG)	Pseudomonadota
*Serratia* sp. AAM11	KX389699	*Serratia marcescens* KRED (AB061685)/96.8	(MG)	Pseudomonadota
*Serratia* sp. AAM12	KX389673	*Serratia marcescens* IARI-UPS 20 (KT441074)/99.0	(MG)	Pseudomonadota
*Bacillus* sp. AAM13	KX389694	*Bacillus subterraneus* DSM13966^T^ (FR733689)/96.7	(MG)	Bacillota
*Leclercia* sp.AAM14	KX389695	*Leclercia adecarboxylata* LMG 2803 (GQ856082)/95.3	(MG)	Pseudomonadota
*Serratia* sp. AAM18	KX827626	*Serratia marcescens* KRED (AB061685)/98.1	(MG)	Pseudomonadota
*Serratia* sp. AAP15	KX827623	*Serratia marcescens* H01-A (AJ297996)/99.6	(HG)	Pseudomonadota
*Enterobacter* sp. AAP16	KX827624	*Enterobacter asburiae* JCM6051^T^ (AB004744)/94.3	(HG)	Pseudomonadota
*Acinetobacter* sp. AAP17	KX827625	*Acinetobacter guillouiae* DSM 590^T^ (X81659)/98.4	(HG)	Pseudomonadota

^a^ Similarity percentage was estimated by considering the number of nucleotide substitutions between a pair of sequences divided by the total number of compared bases × 100%.

**Table 3 insects-16-00711-t003:** Cellulolytic activity and reducing sugar production of bacterial strains isolated from the gut of *A. armida* larvae.

Isolated Gut Section	Strain	Cellulolytic Activity	Clear Zone Diameter (mm)	Reducing Sugars (mg/mL)
Foregut	*Pseudomonas putida* AAT01	(+)	12.3 ± (0.56)	23.895 ± (0.359) *
Foregut	*Enterococcus* sp. AAT02	(−)	ND	ND ^≠^
Foregut	*Enterobacter cloacae* AAT03	(+)	13.1 ± (0.81)	26.031 ± (0.470)
Foregut	*Bacillus* sp. AAT04	(−)	ND	ND
Foregut	*Bacillus* sp. AAT05	(−)	ND	ND
Foregut	*Serratia marcescens* AAT06	(+)	9.6 ± (0.41)	13.934 ± (0.379)
Foregut	*Enterobacter* sp. AAT07	(+)	8.9 ± (0.34)	13.159 ± (0.365)
Midgut	*Serratia marcescens* AAM08	(+)	14.5 ± (0.91)	28.599 ± (0.360)
Midgut	*Serratia* sp. AAM09	(+)	13.8 ± (0.72)	27.397 ± (0.535)
Midgut	*Enterobacter* sp. AAM10	(+)	16.2 ± (0.81)	35.185 ± (0.325)
Midgut	*Serratia* sp. AAM11	(+)	12.9 ± (0.31)	25.621 ± (0.175)
Midgut	*Serratia marcescens AAM*12	(+)	12.3 ± (0.98)	24.781 ± (0.243)
Midgut	*Bacillus* sp. AAM13	(+)	14.0 ± (0.88)	29.540 ± (0.251)
Midgut	*Leclercia* sp. AAM14	(+)	10.1 ± (0.45)	15.816 ± (0.104)
Midgut	*Serratia marcescens* AAC18	(+)	12.1 ± (0.43)	24.780 ± (0.825)
Hindgut	*Serratia marcescens* AAP15	(+)	11.8 ± (0.43)	24.172 ± (0.651)
Hindgut	*Enterobacter asburiae* AAP16	(−)	ND	ND
Hindgut	*Acinetobacter quilouiae* AAP17	(−)	ND	ND

* The values represent the means of the variables obtained from three replicates (with standard deviation in parentheses). Positive activity: (+), Negative activity: (−); ^≠^ ND: Not detected.

## Data Availability

The original contributions presented in this study are included in the article/Appendix A. Further inquiries can be directed to the corresponding author.

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
