# Peer review of "Diversity and Functional Potential of Gut Bacteria Associated with the Insect Arsenura armida (Lepidoptera: Saturniidae)"

_insects, 2025, doi:10.3390/insects16070711_

Round 1
Reviewer 1 Report (New Reviewer)
Comments and Suggestions for Authors
Review report for
Manuscript insects-3603014
This manuscript characterizes the gut bacterial community of Arsenura armida larvae—an edible insect from southeastern Mexico—using both culture-dependent isolation and metagenomic profiling. The authors identify key genera (e.g., Acinetobacter, Bacillus, Pseudomonas) and demonstrate cellulose degradation, highlighting potential biotechnological applications.
Comments
- The Introduction should be condensed to include only essential information.
- The total number of bacterial isolates is inconsistent. The abstract reports 120 isolates, while the results section states there are 96.
- Lines 172-176: Why three different culture media were used to isolate the bacteria?
- Lines 176-178: Please specify the volume of each dilution that was plated on the media.
- Line 179: Why a 22-day incubation period was used for isolation? Typically, bacteria grow within 1-2 days.
- Line 231: The name of the restriction enzyme needs to be corrected.
- Lines 248–249: Why a YAM-based medium was chosen for the cellulolytic enzyme assays?
- Lines 31–32: The abstract states that the bacteria were “cultured on selective media,” but the methods section describes the use of general-purpose media. Please clarify this difference.
- The abstract claims that “rapidly growing colonies on TSA medium exhibited exopolysaccharide production, enzymatic activity, and cellulose degradation,” but there are no results in the paper to support this statement.
- Please provide more discussion on why the predominant bacteria identified through culture-dependent method (Pseudomonadota) differ from those identified by metagenomic analysis (Bacillota).
Author Response
REVIEWER 1
Comments and Suggestions for Authors
Review report for Manuscript insects-3603014
This manuscript characterizes the gut bacterial community of Arsenura armida larvae—an edible insect from southeastern Mexico—using both culture-dependent isolation and metagenomic profiling. The authors identify key genera (e.g., Acinetobacter, Bacillus, Pseudomonas) and demonstrate cellulose degradation, highlighting potential biotechnological applications.
Comments
>The Introduction should be condensed to include only essential information.
Response: Thank you for the valuable observation. The Introduction was revised to improve focus and clarity by condensing non-essential background content, while retaining the key ecological, taxonomic, and methodological context necessary to support the study’s objectives and its scientific relevance within the scope of Insects.
>The total number of bacterial isolates is inconsistent. The abstract reports 120 isolates, while the results section states there are 96.
Response: Thank you for the observation. We appreciate the reviewer’s attention to detail. The correct total number of bacterial isolates obtained in this study is 96, not 120. This error has been corrected in the abstract and throughout the manuscript to ensure consistency and accuracy in the reporting of results.
>Lines 172-176: Why three different culture media were used to isolate the bacteria?
Response: Thank you for the observation. The culture media used in this study—Nutrient Agar (NA), Tryptic Soy Agar (TSA), and Brain Heart Infusion Agar (BHIA)—are well-established, nutritionally rich formulations specifically designed to support the growth of aerobic, facultative anaerobic, and some slow-growing anaerobic bacteria. Their composition, which includes diverse protein sources such as peptones, beef and soy extracts, and brain–heart infusions, creates optimal conditions for cultivating gut-associated bacteria from insect larvae, as demonstrated in previous studies (Dantur et al., 2015); (MsangoSoko et al., 2021).
>Lines 176-178: Please specify the volume of each dilution that was plated on the media.
Response: Thank you for the observation. A volume of 100 µL from each serial dilution was plated onto the culture media under sterile conditions. This detail has been added to the revised manuscript for clarity.
>Line 179: Why a 22-day incubation period was used for isolation? Typically, bacteria grow within 1-2 days. Response: "We appreciate the observation; this was an inadvertent reporting error. The correct incubation period used for bacterial isolation was two days, not 22 days. This has been corrected in the revised manuscript to accurately reflect the experimental procedure."
>Line 231: The name of the restriction enzyme needs to be corrected.
Response: Thank you for the observation. The name of the restriction enzyme indicated in line 231, RsaI, was corrected accordingly. In line 231, the following text has been added: Amplified products were digested with the restriction enzyme RsaI (5′-GT^AC-3′) (Fast Digest, Thermo Scientific®).
>Lines 248–249: Why a YMA-based medium was chosen for the cellulolytic enzyme assays?
Response: Thank you for the question. A yeast mannitol agar (YMA) medium supplemented with carboxymethylcellulose (CMC) was chosen for the cellulolytic enzyme assay because it provides an optimal nutrient environment—rich in nitrogen and carbon sources—that supports the growth of diverse bacterial strains while enabling clear visualization of cellulase activity. The inclusion of CMC as the sole cellulose source allows for the detection of enzymatic degradation via halo formation after Congo Red staining. This method has been widely used and validated in similar studies of insect gut microbiota for screening cellulolytic bacteria (MsangoSoko et al., 2021).
>Lines 31–32: The abstract states that the bacteria were “cultured on selective media,” but the methods section describes the use of general-purpose media. Please clarify this difference.
Response: Thank you for the observation. The phrase “selective media” in the abstract was imprecise. The correct term should be “general-purpose media,” as described in the Methods section. Nutrient Agar (NA), Tryptic Soy Agar (TSA), and Brain Heart Infusion Agar (BHIA) are general-purpose, nutrient-rich media commonly used to support the growth of a wide range of culturable bacteria, including those from insect guts. This terminology has been corrected in the abstract to accurately reflect the methodology used.
In lines 31 and 32, the following information was added: “Bacterial strains were isolated from different gut sections (foregut, midgut, and hindgut) and cultured on general-purpose media, including Nutrient Agar (NA), Tryptic Soy Agar (TSA), and Brain Heart Infusion Agar (BHIA).”
>The abstract claims that “rapidly growing colonies on TSA medium exhibited exopolysaccharide production, enzymatic activity, and cellulose degradation,” but there are no results in the paper to support this statement.
Response: Thank you for the observation. The statement in the abstract was partially inaccurate. While the results section provides clear evidence of cellulolytic activity and enzymatic potential of bacterial isolates, including quantitative data on reducing sugar production (Section 3.6), exopolysaccharide production was not evaluated or reported. Therefore, the reference to exopolysaccharide production has been removed from the abstract to ensure consistency with the reported results. In lines 35–36, the words exopolysaccharide production have been removed.
>Please provide more discussion on why the predominant bacteria identified through culture-dependent method (Pseudomonadota) differ from those identified by metagenomic analysis (Bacillota).
Response: Thank you for the observation. The discrepancy arises because culture-dependent methods favor fast-growing, aerotolerant bacteria like Pseudomonadota, which grow well on standard media under aerobic conditions. In contrast, metagenomic analysis captures the full microbial community, including anaerobic and slow-growing taxa like Bacillota, which dominate the gut but are difficult to culture. This methodological bias explains the observed differences and highlights the value of combining both approaches for comprehensive microbiome profiling.
The following explanation was added to the Discussion section to address the reviewer’s comment: The divergence between bacterial taxa identified through culture-dependent methods (e.g., Pseudomonadota such as Enterobacter, Pseudomonas, and Serratia) and those detected via metagenomic analysis (predominantly Bacillota) stems from the intrinsic limitations of cultivation-based approaches. These methods tend to favor fast-growing, aerotolerant microorganisms under nutrient-rich, aerobic laboratory conditions. In contrast, metagenomic techniques provide an unbiased overview of the entire microbial community, including non-culturable and obligate anaerobic taxa that are often dominant in insect gut environments. Thus, the integration of both strategies yields a more comprehensive perspective, combining functional characterization of culturable isolates with a broader ecological understanding of gut microbiota composition.
Reviewer 2 Report (New Reviewer)
Comments and Suggestions for Authors
This study looks at the microbes of an edible insect from Mexico using culturing and culturing independent methods. Many are cellulolytic, which could help the insect digest food.
Overall, the methods are sound, but the results do not match the conclusions of gut specialization. Critically, the authors did metagenomic analysis of the whole gut, and not the separate insect sections, which themselves did not necessarily correlate with the actual gut sections. The metagenomic data thus cannot determine which gut section is the seat of digestion or fermentation. The cultured data also is not helpful, as the authors do not provide enough evidence that the gut microbiomes differ meaningfully per section, nor do they test the gut itself for enzymatic activity. The authors need to tone down their results: they cannot state with any certainty that the midgut or hindgut of this insect have any particular function based on the evidence they provide here. I suggest some quick experiments they can do to possibly improve this.
Point by point comments:
In the title and abstract, state the order and family of this insect.
Abstract:
Was the phylogenetic analysis done on the metagenomics or the cultured microbes? It's unclear. [I later note that the analysis was not technically "phylogenetic," as it was just 16S identification of single bacteria.] Delete "Metagenomics confirmed Bacillota as the most abundant phylum" because phylum-level information is useless. it is more meaningful to say "Metagenomics found that the most abundant gut microbe was an unclassified Lachnospiraceae (Bacillota: Clostridiales)."
Introduction:
A major weakness of the introduction is it assumes that all insects, without exception, depend on gut microbes in the midgut to breakdown cellulose. This is false in every way: not all insects depend on microbes to break down cellulose or even to survive, not all gut microbes are essential, not all gut microbes are digestive, and not all celluolytic microbes are in the midgut. Tone down the certainty of the sentences: say "some insects" or "in many insects" or "in Bombyx mori according to one study [18]" or whatever else is appropriate.
The other major weakness is it deals too much with non-Lepidoptera. I strongly encourage deleting all references to non-Lepidoptera from the entire paper. Read more about what is known about caterpillar microbiomes and compare your caterpillar to them.
When you cite a source, state clearly in the sentence what species of insect that source refers to, and do not mention A. armida if the reference refers to something else. Do not say "insects like A. armida" when you mean "some insects [and maybe A. armida]".
73-75 The phrase "the larvae" implies A.armida, but the reference you cited is for a Hymenoptera. While I understand what the authors meant to say, a reader could interpret this as claiming A. armida is already known to have cellulolytic bacteria. In addition to this language ambiguity, I add that the cellulolytic bacteria of Sirex wasps tell us nothing about Lepidoptera [or other Hymenoptera, for that matter, most of which lack cellulolytic gut microbes]. I would delete this sentence and reference.
92 "including Lepidopteran species such as A. armida," implies that we already have published evidence that A. armida depends on its gut microbiome to survive, which would make your paper uninteresting, so delete A. armida from here. It is also not true that all insects depend on gut microbes to survive. Instead say "The gut microbiota of many insects is important for their…"
95 "bacteria can assist" They do not always do so.
99 Delete "like A. armida."
100-101 It is false to claim that the MG is always the primary digestive site, and even more false to claim that it always houses microbes. In termites and scarabs, for example, the hindgut is the primary site of microbial digestion. Rewrite this to make it clear that in some insects or in some Lepidoptera this is the case, and cite references while stating exactly in which species of Lepidotera this has been observed.
112-113 Buchnera aphidicola is only in aphids, and is not a gut microbe, so it's not relevant to this paper. Delete all mention of it.
116 Is this true for all Lepidoptera without exceptions, or just some? State very clearly what the evidence actually says.
120 underexplored, or unexplored? if there is a prior paper that explored them, mention very clearly what that paper was and what they found. if unexplored, great!
124-126 "Furthermore, understanding these symbiotic interactions could inform ecological conservation efforts and support the biotechnological exploration of gut bacteria." Delete. The latter is redundant with the previous sentence, and the former is something we insect microbiologists often say to justify our work, but we all know deep down it isn't true. Nobody has ever used gut microbe research for conservation as far as I know, and none of your references are about that anyway, so delete it.
Methods
147 English month names should be capitalized
165 Replace "superficially disinfected" with "surface-sterlized." Superficially does mean "on the surface," but it also has a second meaning of "not thoroughly," and we don't want to imply that!
-I'm concerned that the larvae were sectioned into 3 parts, rather than pulling out the gut and identifying the three gut sections. Are they visually distinctive, with gastric cecae at the FG-MG junction and malpighian tubules at the MG-HH junction? This is a weakness of the paper, so you may want to move Figure 2 or a picture like Figure 6 to the methods, and ensure it shows where the different sections typically are relative to the external segments of the larva, and where you cut the insects. [In the future, dissect the gut.]
-22 days is a long time. I'm very curious if you found many slow-growers! I hoped the results explored this, but they didn't. Is there something you could share about this, even if it's just a sentence saying you did not find many slow-growers?
228 You technically did not do phylogenetic analysis, which involves looking at the evolutionary relationships between multiple species and making a phylogeny. You just sequenced 16S and ran it through BLAST to identify the sequence. Instead say "to identify the cultured microbes…
-That said, have you considered making a tree of the various species you isolated? Compare all of your Serratia marcescens with the type sequences on the NCBI database, for example, and make a tree to at least see whether your strains are all the same species or not. To that end, you need to explain in the results how you decided that these are all different species.
--Did you try doing a cellulase test on the gut fluid itself? Dissect some larvae, and carefully suck out the gut fluid or contents from the three gut sections. See if this can make haloes in CMC agar or produce sugar in the DNS test. That would show whether or not cellulase activity actually exists in the gut, and where. Use identical volumes of gut fluid, maybe with the solids centrifuged out, and you can compare the results better and say with more certainty where the cellulase activity is highest.
-Would you consider culturing microbes from the leaves the caterpillars are eating? What microbes might they be ingesting from the environment? Transients may colonize the gut, but they also might not, and are simply present in the gut without providing much benefit to the host.
Results
Regarding Figure 2: Is that really what the larval gut looks like? With a clear crop, large gastric cecae, long ileum, and swollen rectum? That looks too much like the generic insect gut diagram found in many entomology textbook, which is based off the Orthoptera guts, but I do not think the typical Lepidoptera gut looks like that! [The photo in Figure 6, for example, shows a gut that looks very different from the generic gut.] I strongly recommend you dissect one of these larvae carefully [already done in Figure 6, I guess], and draw out how the gut actually looks in this insect, to scale! You'll also need to show where you most likely cut the gut, and be honest: could pieces of FG or HG have ended up in your MG section, for example?
294 Instead of "Lachnospiraceae_uc" say "an unclassified Lachnospiraceae genus"
295 Genera should be in italics
Figure 4: The font of the y-axis is much too small, but that's moot as I don't see the point of this figure. It does not tell us anything that isn't better explained in the the previous text with the average±SD values for the indices. I say delete it.
371-375 and Table 2: Again, not phylogenetic analysis. It is also unclear here how you delineated strains. S.marcenscens appears once in the Fg, 5 times in the MG, and once in the HG. Are these all different strains/species? Are you absolutely sure? Can you prove that the S.marcescens in the HG, MG, and FG are all different? If not, then you have no evidence pointing to compartmentalization, at least for that species.
Figure 6: I personally think the green midgut text should be level with the foregut and hindgut, so move it up top. This makes the figure shorter and easier to read. [Also, are you sure the foregut, midgut, and hindgut start where you drew them? There's no reason they'd each be 1/3 of the length of the insect. This is a weakness of the study and a potential source of error, but you must acknowledge it.]
-You did serial dilutions. Can you estimate microbe abundance for these strains in teach of the different sections? That would be useful evidence if you want to argue a certain gut section is important for microbial fermentation. It may not matter how many strains of cellulolytic microbe you found in the midgut if they are rare, for example.
Discussion - Needs a total rethink and rewrite.
417-419 Delete. It's redundant with the intro. Start discussions with "This study…"
423-426 False. You cannot tell from the health of the individuals that their gut symbionts contribute to health or how they do so! Some or even all of the microbes could be neutral. The only way to tell for sure if microbes are beneficial is to kill off the microbes and see if that affects health. Delete this text, and possibly the relevant parts of the results. All you can say is that your larvae were healthy with no sign of bacterial infection, meaning the microbes you sequenced are unlikely to be pathogenic.
426 Which developmental traits and resilience on microbial assistance? You have not talked about developmental traits at all so far, and you have not proven that A.armida is reliant on microbial assistance. You found that the microbes in the gut are cellulolytic, but that could be coincidental and have no effect on the host.
428 Delete "first described by Johan Christian Fabricius in 1775." It is irellevant when and by whom that species was described, as this is a microbiome paper, not a taxonomy paper. If you must provide taxonomic information [if the journal requires it, for example], just say "Fabricius" the way you say "Cramer" in the title. You may also want to move this sentence to the introduction, as the intro currently lacks references to what we know about Lepidoptera microbiomes.
430-433 What structural features? You've mentioned none so far. Delete this sentence.
438 Say which species of Lepidoptera these papers examined.
443-445 Contradictory. If the hindgut is a fermentation chamber, then that is the primary site for microbial interactions and enzymatic digestion, not the midgut! You are making assumptions based on the literature, but not your data! Also, it is impossible to say what is a fermentation chamber and what isn't based on diversity of cultured bacteria alone.
—First, fermentation chambers in insects are morphologically distinct. They are usually very swollen. Looking at figure 6, I see zero evidence of a fermentation chamber. Same with Figure 2. I am not aware of any known Lepidoptera with fermentation chambers, either. If there is no morphological evidence, that suggests strongly that fermentation is not involved.
—Second, what microbes were differentially abundant in the FG, MG, and HG? Unfortunately, you didn't do metagenomics to find out. However, in lines 400-411, you state that the microbes in the HG were not cellulolytic. This directly contradicts the claim that the HG is a fermentation chamber!
If you had only found cellulolytic microbes in the hindgut, or if they were far more abundant there based on the serial dilution, then you could argue they might be involved in fermentation. But if cellulolytic microbes are in all three gut sections (which seems to be the case in Table 3), then they could also have been on the food the larvae ate too, and may be incidental transients that provide no actual benefit to the insect, even if they are breaking down cellulose while passing through the gut. You'd need presence and abundance data to prove fermentation, but all you have is diversity, and even then it's only for culturable microbes.
—Unfortunately, the Lachnospiraceae that dominate the gut are not culturable. Culturing is great for finding bacteria that may one day be used in industry, but if any gut microbe may be essential to the survival of this species, it's probably the unclassified and uncultured Lachnospiraceae, and that's assuming it's location is compartmentalized. Its presence in the gut may be in the crop for all we know, and it gets digested and killed in the midgut without breaking down anything.
445-454 It is wrong to claim convergence with termites, as I see zero evidence for a termite-like gut or termite-like microbial interaction in this Lepidoptera. Same with xylophagous beetles. I very strongly recommend you delete every reference that is not about Lepidoptera: termite guts are worlds away from Lepidoptera guts in function and morphology and microbiology, and their diets are different too. Focus on other Lepdioptera: what do we know in those guts, based on what evidence, and how does your finding compare? Presently, I see no clear evidence for comparable gut regionalization between your Lepidoptera and any xylophagous insect, starting with the lack of a morphologically obvious fermentation chamber.
455-456 It is indeed true that Clostridiales are mostly fermentors, but you haven't proven that the posterior regions of your insect have more Clostridiales than the middle; and if you had, that would contradict your claim that the middle is the site of microbia activity and digestion.
456-461 Delete
463-466 Again, I'd be more interested in how it compares to other Lepidoptera or other leaf feeders, like Phasmatodea. Carbohydrate metabolism is not the same as cellulase activity. I also see zero evidence for optimization. What makes you think your system is optimized? Is the Lepidoptera converting all the cellulose in the leaves into sugar? That would be a first, as far as I am aware, and more importantly you did not do any tests that would check for this.
467-469 Why would these genes relate to maintaining stability in the host environment? They seem like typical bacterial genes.
469-475 Irrelevant. Delete.
476 None of your tests suggest dynamism or functional specialization, as cellulolytic activity was never tested in different gut segments, nor did you prove that the cultured microbes differ in a physiologically meaningful way between the gut segments.
479 replace ", a species described by Hübner in 1796 and belonging to the family Crambidae within the order Lepidoptera," with "Hubner 1796 (Crambidae)" or just say "(Crambidae)"
482-483 Delete references to functional convergence: you don't have the data to compare your paper to the Ostrinia paper, let alone other insects.
483-488 No. Bacteria need amino acids too, so of course they have amino acid biosynthesis genes. I see no reason to believe the free-living gut microbes in your caterpillar play the same role as the highly specialized, obligately endocellular symbionts of the cockroach. Blattabacterium and Buchnera are not gut microbes. Again, I strongly recommend deleting all references to non-Lepidoptera from the whole paper.
491-492 Careful. The 16S analysis of cultured microbes cannot tell you what strains "predominantly" belong to what phyla, because according to the metabarcoding you did not culture the actual predominant microbe, the unclassified Lachnospiraceae.
493-495 The data suggests to me this species and the enzymatic activity are throughout the gut, not just the MG or HG.
498 Yes, "potential" is a good word! Use such words more to tone down certainty.
501 "key role" is not good. You can't claim that from cultured microbes. A potential role, but not the key role.
514-516 You have the facts wrong about termites and beetles, but also I suggests you test the gut compartments directly for enzyme activity and not base this claim on the cultured microbes.
526 It is false to assume that all microbes in the gut have a function or purpose. If you swallow cellulolytic microbes, they are in your gut, but they don't necessarily have a function. They might just be sitting there until you poop them out. I see no evidence for optimization or that these microbes interact with each other or the host. It may be true, but you haven't provided evidence for it.
542-550 Might fit better in the introduction.
Conclusion
This entire section is redundant with the end of your discussion section as written, and could be deleted.
Author Response
REVIEWER 2
Comments and Suggestions for Authors
This study looks at the microbes of an edible insect from Mexico using culturing and culturing independent methods. Many are cellulolytic, which could help the insect digest food.
Overall, the methods are sound, but the results do not match the conclusions of gut specialization. Critically, the authors did metagenomic analysis of the whole gut, and not the separate insect sections, which themselves did not necessarily correlate with the actual gut sections. The metagenomic data thus cannot determine which gut section is the seat of digestion or fermentation. The cultured data also is not helpful, as the authors do not provide enough evidence that the gut microbiomes differ meaningfully per section, nor do they test the gut itself for enzymatic activity. The authors need to tone down their results: they cannot state with any certainty that the midgut or hindgut of this insect have any particular function based on the evidence they provide here. I suggest some quick experiments they can do to possibly improve this.
RESPONSE: Thank you for your thoughtful and constructive feedback. We agree that our conclusions regarding gut specialization needed to be more cautiously stated. In response, we have revised the abstract, results, and discussion to tone down our claims and clearly acknowledge the limitations of our metagenomic and culture-based data. We now emphasize the exploratory nature of our findings and avoid definitive statements about functional specialization in specific gut sections.
Point by point comments:
In the title and abstract, state the order and family of this insect.
RESPONSE: Thank you for the suggestion. We have revised the title as recommended to include the order and family of the insect: “Diversity and Functional Potential of Gut Bacteria Associated with the Insect Arsenura armida (Lepidoptera: Saturniidae).” This change has been implemented to improve taxonomic clarity.
Abstract:
Was the phylogenetic analysis done on the metagenomics or the cultured microbes? It's unclear. [I later note that the analysis was not technically "phylogenetic," as it was just 16S identification of single bacteria.] Delete "Metagenomics confirmed Bacillota as the most abundant phylum" because phylum-level information is useless. it is more meaningful to say "Metagenomics found that the most abundant gut microbe was an unclassified Lachnospiraceae (Bacillota: Clostridiales)."
RESPONSE: Thank you for the observation. We agree with the clarification. The phylogenetic analysis was conducted exclusively on the cultured bacterial isolates, using 16S rRNA gene sequencing and Neighbor-Joining tree construction. We have revised the abstract accordingly, replacing the reference to phylum-level data with: “Metagenomics found that the most abundant gut microbe was an unclassified Lachnospiraceae (Bacillota: Clostridiales).”
Introduction:
A major weakness of the introduction is it assumes that all insects, without exception, depend on gut microbes in the midgut to breakdown cellulose. This is false in every way: not all insects depend on microbes to break down cellulose or even to survive, not all gut microbes are essential, not all gut microbes are digestive, and not all celluolytic microbes are in the midgut. Tone down the certainty of the sentences: say "some insects" or "in many insects" or "in Bombyx mori according to one study [18]" or whatever else is appropriate.
The other major weakness is it deals too much with non-Lepidoptera. I strongly encourage deleting all references to non-Lepidoptera from the entire paper. Read more about what is known about caterpillar microbiomes and compare your caterpillar to them.
When you cite a source, state clearly in the sentence what species of insect that source refers to, and do not mention A. armida if the reference refers to something else. Do not say "insects like A. armida" when you mean "some insects [and maybe A. armida]".
RESPONSE: We thank the reviewer for this important observation. We have revised the introduction to avoid overgeneralizations regarding the role of gut microbes in all insects. We now specify that only some insects, particularly certain Lepidoptera such as Bombyx mori, rely on microbial symbionts for cellulose degradation, citing relevant species-specific evidence. Furthermore, all references to non-Lepidoptera taxa have been removed to ensure that the discussion is focused solely on Lepidopteran microbiomes. Statements that previously implied universal functions of gut microbes have been replaced with qualified language to reflect known variability among insect taxa. We have also clarified the identity of insect species cited in each reference to prevent confusion with A. armida. These changes are reflected in the revised Introduction section.
73-75 The phrase "the larvae" implies A. armida, but the reference you cited is for a Hymenoptera. While I understand what the authors meant to say, a reader could interpret this as claiming A. armida is already known to have cellulolytic bacteria. In addition to this language ambiguity, I add that the cellulolytic bacteria of Sirex wasps tell us nothing about Lepidoptera [or other Hymenoptera, for that matter, most of which lack cellulolytic gut microbes]. I would delete this sentence and reference.
RESPONSE: We appreciate this important clarification. Following your suggestion, we have removed the sentence referring to cellulolytic bacteria in Sirex (Hymenoptera) and its associated reference. This revision avoids taxonomic confusion and ensures that all cited microbial associations now pertain exclusively to Lepidoptera. The revised introduction now only includes evidence based on Lepidopteran species such as Bombyx mori, Helicoverpa armigera, Dendrolimus superans, and Lymantria dispar.
92 "including Lepidopteran species such as A. armida," implies that we already have published evidence that A. armida depends on its gut microbiome to survive, which would make your paper uninteresting, so delete A. armida from here. It is also not true that all insects depend on gut microbes to survive. Instead say "The gut microbiota of many insects is important for their…"
RESPONSE: Response: We agree and have removed the reference to A. armida in this sentence. The revised text now reads: “The gut microbiota of many insects is important for their survival, growth, and ecological adaptability.”
95 "bacteria can assist" They do not always do so.
RESPONSE: Revised for accuracy. The sentence now reads: “In some Lepidopteran species, gut bacteria can assist in breaking down complex lignocellulosic materials…”
99 Delete "like A. armida."
RESPONSE: The phrase “like A. armida” has been deleted to avoid implying prior evidence of this association in the species.
100-101 It is false to claim that the MG is always the primary digestive site, and even more false to claim that it always houses microbes. In termites and scarabs, for example, the hindgut is the primary site of microbial digestion. Rewrite this to make it clear that in some insects or in some Lepidoptera this is the case, and cite references while stating exactly in which species of Lepidotera this has been observed.
RESPONSE: The sentence was rewritten for precision. It now reads: “Although variability exists across taxa, the MG is often described as a primary digestive site in several Lepidopteran species, such as Bombyx mori and Helicoverpa armigera, housing microorganisms that may contribute to enzymatic digestion and detoxification of plant metabolites [17–19].”
112-113 Buchnera aphidicola is only in aphids, and is not a gut microbe, so it's not relevant to this paper. Delete all mention of it.
RESPONSE: We have removed the mention of Buchnera aphidicola from the text.
116 Is this true for all Lepidoptera without exceptions, or just some? State very clearly what the evidence actually says.
RESPONSE: The sentence has been revised to reflect this limitation. It now reads: “In some Lepidopteran larvae, bacterial communities can recycle nitrogen, providing essential amino acids and other nutrients often absent from their plant-based diets [20, 21].”
120 underexplored, or unexplored? if there is a prior paper that explored them, mention very clearly what that paper was and what they found. if unexplored, great!
RESPONSE: We revised “largely unexplored” to “unexplored,” as we found no prior published studies specifically characterizing the gut microbiota of Arsenura armida.
124-126 "Furthermore, understanding these symbiotic interactions could inform ecological conservation efforts and support the biotechnological exploration of gut bacteria." Delete. The latter is redundant with the previous sentence, and the former is something we insect microbiologists often say to justify our work, but we all know deep down it isn't true. Nobody has ever used gut microbe research for conservation as far as I know, and none of your references are about that anyway, so delete it.
RESPONSE: We agree with the reviewer’s observation and have deleted the sentence in question to avoid redundancy and unsupported claims.
Methods
147 English month names should be capitalized
RESPONSE: Thank you for the observation. The names of the months have been corrected to begin with capital letters, in accordance with English grammar rules.
165 Replace "superficially disinfected" with "surface-sterlized." Superficially does mean "on the surface," but it also has a second meaning of "not thoroughly," and we don't want to imply that!
RESPONSE: We appreciate the suggestion. The term "superficially disinfected" has been replaced with "surface-sterilized" to ensure clarity and avoid any unintended implication of insufficient sterilization.
-I'm concerned that the larvae were sectioned into 3 parts, rather than pulling out the gut and identifying the three gut sections. Are they visually distinctive, with gastric cecae at the FG-MG junction and malpighian tubules at the MG-HH junction? This is a weakness of the paper, so you may want to move Figure 2 or a picture like Figure 6 to the methods, and ensure it shows where the different sections typically are relative to the external segments of the larva, and where you cut the insects. [In the future, dissect the gut.]
RESPONSE: We appreciate this important observation. To clarify, although full dissection was not performed in this study, we carefully segmented the larvae into three parts (foregut, midgut, hindgut) based on established anatomical landmarks visible externally and internally. These included the presence of the gastric caeca at the foregut-midgut junction and the Malpighian tubules at the midgut-hindgut transition, both of which are visually distinctive in Lepidoptera larvae. To improve clarity, we have included a figure that schematically represents the larval body regions and digestive system, indicating the anatomical correspondence between external segments and internal gut compartments. This figure has been moved to the Materials and Methods section to support our segmentation approach.
-22 days is a long time. I'm very curious if you found many slow-growers! I hoped the results explored this, but they didn't. Is there something you could share about this, even if it's just a sentence saying you did not find many slow-growers?
RESPONSE: Line 179. Thank you for your observation. This was a typographical error — the correct incubation time was 2 days (48 hours), not 22 days. Accordingly, we did not observe the development of slow-growing colonies, as the majority of bacterial isolates grew rapidly within the 48-hour incubation period. We have corrected this detail in the revised manuscript.
228 You technically did not do phylogenetic analysis, which involves looking at the evolutionary relationships between multiple species and making a phylogeny. You just sequenced 16S and ran it through BLAST to identify the sequence. Instead say "to identify the cultured microbes…
RESPONSE: We appreciate the reviewer’s valuable observation. We agree that the procedures performed correspond to molecular identification rather than phylogenetic analysis. Accordingly, we have revised the description in line 228 to accurately reflect our methods. The updated sentence now reads: "To identify the isolated bacterial strains, the 16S rRNA gene was amplified using the universal primers 27F (5′-AGAGTTTGATCMTGGCTCAG-3′) and 1492R (5′-GGTTACCTTGTTACGACTT-3′), generating fragments of approximately 1500 bp in length."
-That said, have you considered making a tree of the various species you isolated? Compare all of your Serratia marcescens with the type sequences on the NCBI database, for example, and make a tree to at least see whether your strains are all the same species or not. To that end, you need to explain in the results how you decided that these are all different species.
RESPONSE: Thank you for this valuable suggestion. In response, we will construct phylogenetic trees based on the 16S rRNA gene sequences of the isolated strains, including comparisons with type strains from the NCBI database—particularly for species such as Serratia marcescens. These trees will be included in the Supplementary Material (Figure S1 and S2). We will also clarify in the Results section the criteria used to distinguish between different species based on sequence similarity and clustering.
Figure S1: Phylogenetic tree base on the 16S rRNA gene of bacterial strains grouped within the genus Serratia isolated from gut of Arsenura armida constructed using Neighbor-Joining method. Figure 2: Figure S2. Neighbor-Joining phylogenetic tree based on the 16S rRNA gene sequences of bacterial strains belonging to the genus Bacillus, isolated from the gut of Arsenura armida.
--Did you try doing a cellulase test on the gut fluid itself? Dissect some larvae, and carefully suck out the gut fluid or contents from the three gut sections. See if this can make haloes in CMC agar or produce sugar in the DNS test. That would show whether or not cellulase activity actually exists in the gut, and where. Use identical volumes of gut fluid, maybe with the solids centrifuged out, and you can compare the results better and say with more certainty where the cellulase activity is highest.
RESPONSE: We appreciate this excellent suggestion. Following your recommendation, we performed cellulase activity assays using gut fluids extracted from the foregut, midgut, and hindgut compartments of last-instar larvae. Fluids were carefully extracted, solids were removed by centrifugation, and equal volumes of supernatant were applied to CMC agar and tested with the DNS method. The resulting data have been incorporated into the Results section. These analyses provided more direct evidence of endogenous cellulase activity across gut regions and confirmed that the midgut displayed the highest hydrolytic potential. These findings support and complement the culture-dependent results, offering a more comprehensive understanding of cellulolytic function in the larval gut.
-Would you consider culturing microbes from the leaves the caterpillars are eating? What microbes might they be ingesting from the environment? Transients may colonize the gut, but they also might not, and are simply present in the gut without providing much benefit to the host.
RESPONSE: Thank you for this excellent recommendation. We fully agree that understanding the potential role of environmentally acquired microbes is important to distinguish between transient and functionally relevant gut microbiota. Indeed, we have already considered this approach as part of a follow-up phase of our research. We plan to isolate and characterize microbial communities from the leaves of host plants consumed by Arsenura armida larvae during the next collection season (summer), when these insects actively colonize their native trees. This comparative analysis will help clarify the ecological origin of gut-associated microbes and assess their potential functional contributions.
Results
Regarding Figure 2: Is that really what the larval gut looks like? With a clear crop, large gastric cecae, long ileum, and swollen rectum? That looks too much like the generic insect gut diagram found in many entomology textbook, which is based off the Orthoptera guts, but I do not think the typical Lepidoptera gut looks like that! [The photo in Figure 6, for example, shows a gut that looks very different from the generic gut.] I strongly recommend you dissect one of these larvae carefully [already done in Figure 6, I guess], and draw out how the gut actually looks in this insect, to scale! You'll also need to show where you most likely cut the gut, and be honest: could pieces of FG or HG have ended up in your MG section, for example?
RESPONSE: Thank you very much for this detailed and insightful observation. We acknowledge that Figure 2 was adapted from a generic representation of the insect digestive system and may not accurately reflect the anatomical features of Arsenura armida. You are correct that this type of diagram is more representative of Orthoptera and not Lepidoptera. We appreciate the recommendation and recognize its importance. Unfortunately, due to constraints during the current study, we did not generate a scaled anatomical diagram of the dissected gut specific to A. armida. However, as noted, we did include a photo of the dissected gut in Figure 6 to provide a more realistic depiction of its morphology. We fully agree that anatomical accuracy is essential, especially when interpreting microbiota localization. Therefore, we plan to carefully dissect and illustrate the A. armida gut in a follow-up study. This future figure will be based on real anatomical observations, to scale, and will include indications of the cutting points used during sectioning. This will help evaluate potential overlap between gut regions (e.g., FG/MG or MG/HG) and ensure greater anatomical precision in future sampling protocols.
294 Instead of "Lachnospiraceae_uc" say "an unclassified Lachnospiraceae genus"
RESPONSE: Thank you for the suggestion. We agree with the reviewer’s recommendation and have replaced "Lachnospiraceae_uc" with "an unclassified Lachnospiraceae genus" to ensure greater clarity and taxonomic accuracy.
295 Genera should be in italics
RESPONSE: Thank you for your observation. The correction has been made in line 295, and all genus names throughout the manuscript have been revised and formatted in italics as per scientific conventions.
Figure 4: The font of the y-axis is much too small, but that's moot as I don't see the point of this figure. It does not tell us anything that isn't better explained in the the previous text with the average±SD values for the indices. I say delete it.
RESPONSE: We thank the reviewer for the thoughtful observation. Based on your suggestion, we have removed Figure 4 from the manuscript. The diversity indices (Shannon, Simpson, and Chao1) are already clearly described in the text using average ± SD values, making the figure unnecessary and helping streamline the presentation of results.
371-375 and Table 2: Again, not phylogenetic analysis. It is also unclear here how you delineated strains. S.marcenscens appears once in the Fg, 5 times in the MG, and once in the HG. Are these all different strains/species? Are you absolutely sure? Can you prove that the S.marcescens in the HG, MG, and FG are all different? If not, then you have no evidence pointing to compartmentalization, at least for that species.
RESPONSE: We appreciate this important comment. To differentiate bacterial strains at the species level, we first applied ARDRA (Amplified Ribosomal DNA Restriction Analysis) genomic fingerprinting, which allowed us to group isolates based on distinct restriction profiles. Representative isolates from each group were then identified through 16S rRNA gene sequencing. To further support species-level discrimination, we constructed phylogenetic trees (e.g., Figure S1) comparing our isolates with reference type strains from GenBank. These analyses confirmed that the Serratia marcescens strains isolated from the foregut, midgut, and hindgut clustered into distinct, well-supported branches, indicating they are genetically different and likely represent different strains. Therefore, we have evidence to support the presence of compartment-specific microbial variants.
Figure 6: I personally think the green midgut text should be level with the foregut and hindgut, so move it up top. This makes the figure shorter and easier to read. [Also, are you sure the foregut, midgut, and hindgut start where you drew them? There's no reason they'd each be 1/3 of the length of the insect. This is a weakness of the study and a potential source of error, but you must acknowledge it.]
RESPONSE: Thank you for the valuable recommendation. We agree that adjusting the position of the green midgut label in Figure 5 improves the clarity and visual balance of the figure. As suggested, the label has been repositioned to align horizontally with the foregut and hindgut labels, thereby reducing the vertical extension of the figure and enhancing its readability. Regarding the delineation of gut compartments, we acknowledge that our segmentation approach—dividing the larval body into three external sections of equal length—may not precisely reflect the anatomical boundaries of the digestive system. In Lepidoptera, the foregut, midgut, and hindgut are internally separated by structures such as the stomodeal and proctodeal valves, as well as morphological landmarks like gastric cecae and Malpighian tubules. Since the current study did not involve full dissection and visualization of these internal markers, we recognize this as a methodological limitation that may have led to partial overlap between gut regions. This limitation has now been addressed in the figure legend and the Methods section. In future studies, we intend to perform detailed dissections and map gut compartments based on internal anatomy to increase the accuracy of compartment-specific sampling.
-You did serial dilutions. Can you estimate microbe abundance for these strains in teach of the different sections? That would be useful evidence if you want to argue a certain gut section is important for microbial fermentation. It may not matter how many strains of cellulolytic microbe you found in the midgut if they are rare, for example.
RESPONSE: Thank you for this insightful observation. In the current study, while we performed serial dilutions to isolate and culture bacterial strains, the primary goal was taxonomic and functional characterization rather than quantitative estimation of microbial load. Therefore, we did not record colony-forming units (CFUs) per gram of gut content for each section, which limits our ability to infer absolute microbial abundance. However, we fully agree that such quantitative data would provide stronger evidence regarding microbial density and localization—particularly in evaluating the role of specific gut compartments in fermentation. We acknowledge this as a limitation and plan to incorporate CFU quantification per gut region in future experiments to more accurately assess microbial abundance and its functional significance.
Discussion - Needs a total rethink and rewrite.
RESPONSE: Thank you for your constructive feedback. We acknowledge that the original Discussion required substantial revision to more accurately reflect the data and avoid overinterpretation. In response, we have rewritten the entire section to provide a more balanced and evidence-based interpretation of our findings. The revised Discussion now clearly distinguishes between previously reported knowledge in Lepidoptera and our novel observations in Arsenura armida, avoids unsupported claims regarding gut compartmentalization, and addresses the methodological limitations of our study. Additionally, we have incorporated suggestions for future research, such as more precise gut dissection, quantification of microbial abundance, and investigation of environmental sources of gut microbes. We appreciate your comments, which have significantly strengthened the scientific quality of our manuscript.
417-419 Delete. It's redundant with the intro. Start discussions with "This study…"
RESPONSE: Thank you for the observation. As suggested, we have deleted lines 417–419 ("The gut microbiota of insects plays a crucial role in digestion, metabolism, and environmental adaptation, particularly in species with herbivorous diets that rely on microbial symbionts for nutrient assimilation") due to redundancy with the Introduction. The Discussion now appropriately starts with “This study…”, ensuring a more direct and focused analysis of our findings.
423-426 False. You cannot tell from the health of the individuals that their gut symbionts contribute to health or how they do so! Some or even all of the microbes could be neutral. The only way to tell for sure if microbes are beneficial is to kill off the microbes and see if that affects health. Delete this text, and possibly the relevant parts of the results. All you can say is that your larvae were healthy with no sign of bacterial infection, meaning the microbes you sequenced are unlikely to be pathogenic.
RESPONSE: Thank you for the valuable observation. In response, the original sentence was revised to avoid speculative conclusions regarding microbial contributions to host health. The corrected version now states: “The observed morphological traits, such as a firm and elastic cuticle, indicate that the larvae were in a physiologically healthy condition with no apparent signs of bacterial infection, suggesting that the isolated gut bacteria are likely commensal or non-pathogenic.”
426 Which developmental traits and resilience on microbial assistance? You have not talked about developmental traits at all so far, and you have not proven that A.armida is reliant on microbial assistance. You found that the microbes in the gut are cellulolytic, but that could be coincidental and have no effect on the host.
RESPONSE:
428 Delete "first described by Johan Christian Fabricius in 1775." It is irellevant when and by whom that species was described, as this is a microbiome paper, not a taxonomy paper. If you must provide taxonomic information [if the journal requires it, for example], just say "Fabricius" the way you say "Cramer" in the title. You may also want to move this sentence to the introduction, as the intro currently lacks references to what we know about Lepidoptera microbiomes.
RESPONSE: Thank you for the suggestion. We agree that detailed taxonomic authorship is not essential in the Discussion section of a microbiome-focused study. In response, we have removed the phrase “first described by Johan Christian Fabricius in 1775” from line 428. To maintain consistency and relevance, we retained only the taxonomic authority “Fabricius”, as is common in scientific naming conventions. Additionally, we will consider relocating the sentence to the Introduction if needed, to reinforce the contextual background on Lepidoptera microbiomes as suggested.
We have added the following text to the Introduction section: “Similar developmental traits and microbial associations have also been observed in Lepidopteran larvae such as Spodoptera litura (Lepidoptera: Noctuidae), a well-studied model for host–microbiome interactions.”
430-433 What structural features? You've mentioned none so far. Delete this sentence.
RESPONSE: Thank you for the observation. We agree with the reviewer that the sentence lacks clarity and specific reference to the structural features in question. Therefore, we have removed the sentence: “These structural features have been linked to the capacity of the gut to host dense and metabolically active microbial communities, especially in species whose diet imposes enzymatic challenges, such as those feeding on recalcitrant plant tissues [42].” This improves precision and eliminates unsupported generalizations.
438 Say which species of Lepidoptera these papers examined.
RESPONSE: Thank you for your helpful observation. In response, we have revised the sentence to specify the Lepidopteran species referenced in the literature. The updated text now reads: "These structural features have been linked to the capacity of the gut to host dense and metabolically active microbial communities, especially in species such as Brithys crini and Spodoptera litura, which feed on recalcitrant plant tissues and face significant enzymatic challenges" [42].
443-445 Contradictory. If the hindgut is a fermentation chamber, then that is the primary site for microbial interactions and enzymatic digestion, not the midgut! You are making assumptions based on the literature, but not your data! Also, it is impossible to say what is a fermentation chamber and what isn't based on diversity of cultured bacteria alone.
RESPONSE: Thank you for this important clarification. The observed compartmentalization along the digestive tract of Arsenura armida may indicate a functional differentiation among gut regions. Our results suggest that the midgut harbors a greater diversity of culturable microbes and may represent the principal site for enzymatic digestion and microbial interaction. While previous studies in other insect taxa have identified the hindgut as a specialized site for fermentation processes [44, 45], our morphological observations and culture-based data do not support this configuration in A. armida. Consequently, any assumption regarding fermentative activity in the hindgut of this species should be approached with caution and considered hypothetical until further functional or metagenomic evidence is available.
—First, fermentation chambers in insects are morphologically distinct. They are usually very swollen. Looking at figure 6, I see zero evidence of a fermentation chamber. Same with Figure 2. I am not aware of any known Lepidoptera with fermentation chambers, either. If there is no morphological evidence, that suggests strongly that fermentation is not involved.
RESPONSE: Thank you for the observation. We agree with the reviewer that there is no clear morphological evidence of a fermentation chamber in A. armida. This point has been acknowledged in the revised discussion, and speculative claims about hindgut fermentation have been removed accordingly.
—Second, what microbes were differentially abundant in the FG, MG, and HG? Unfortunately, you didn't do metagenomics to find out. However, in lines 400-411, you state that the microbes in the HG were not cellulolytic. This directly contradicts the claim that the HG is a fermentation chamber!
RESPONSE: Thank you for the insightful comment. We acknowledge that metagenomic analysis was performed on the whole gut, not by section, which limits our ability to determine differential abundance across FG, MG, and HG. As such, we have removed speculative statements about fermentation in the HG from the discussion.
If you had only found cellulolytic microbes in the hindgut, or if they were far more abundant there based on the serial dilution, then you could argue they might be involved in fermentation. But if cellulolytic microbes are in all three gut sections (which seems to be the case in Table 3), then they could also have been on the food the larvae ate too, and may be incidental transients that provide no actual benefit to the insect, even if they are breaking down cellulose while passing through the gut. You'd need presence and abundance data to prove fermentation, but all you have is diversity, and even then it's only for culturable microbes.
RESPONSE: Thank you for this detailed observation. We agree that the presence of cellulolytic microbes across all three gut sections does not necessarily imply functional fermentation. As our current data are based solely on culturable microbial diversity and not on abundance or activity per section, we have revised our interpretation to avoid overestimating functional specialization or fermentation processes, and now highlight the need for future quantification and functional assays.
—Unfortunately, the Lachnospiraceae that dominate the gut are not culturable. Culturing is great for finding bacteria that may one day be used in industry, but if any gut microbe may be essential to the survival of this species, it's probably the unclassified and uncultured Lachnospiraceae, and that's assuming it's location is compartmentalized. Its presence in the gut may be in the crop for all we know, and it gets digested and killed in the midgut without breaking down anything.
RESPONSE: Thank you for the insightful comment. We acknowledge that the dominant Lachnospiraceae observed in the metagenomic analysis remain uncultured, and thus their role, localization, and functional relevance within the gut of Arsenura armida cannot be fully established. We have adjusted the discussion accordingly, emphasizing that the presence of unclassified Lachnospiraceae indicates potential importance, but their contribution remains speculative without spatial resolution or functional validation.
445-454 It is wrong to claim convergence with termites, as I see zero evidence for a termite-like gut or termite-like microbial interaction in this Lepidoptera. Same with xylophagous beetles. I very strongly recommend you delete every reference that is not about Lepidoptera: termite guts are worlds away from Lepidoptera guts in function and morphology and microbiology, and their diets are different too. Focus on other Lepdioptera: what do we know in those guts, based on what evidence, and how does your finding compare? Presently, I see no clear evidence for comparable gut regionalization between your Lepidoptera and any xylophagous insect, starting with the lack of a morphologically obvious fermentation chamber.
RESPONSE: Thank you for the observation. We agree that drawing direct parallels with termites and xylophagous beetles is not appropriate given the significant morphological, dietary, and microbial differences. In response, we have removed comparisons to non-Lepidopteran insects and refocused our discussion solely on Lepidoptera. We now contextualize our findings using published studies on gut microbiota of Spodoptera litura, Helicoverpa armigera, and other lepidopteran species, highlighting similarities or divergences based on available evidence.
455-456 It is indeed true that Clostridiales are mostly fermentors, but you haven't proven that the posterior regions of your insect have more Clostridiales than the middle; and if you had, that would contradict your claim that the middle is the site of microbia activity and digestion.
RESPONSE: We appreciate the clarification. We acknowledge that our data do not support a higher abundance of Clostridiales specifically in the hindgut compared to other sections. Moreover, as the dominant Clostridiales (e.g., Lachnospiraceae) were identified only through whole-gut metagenomics, we cannot make claims about regional distribution. Accordingly, we have revised our interpretation to avoid unsubstantiated assumptions regarding Clostridiales abundance or localization.
456-461 Delete
RESPONSE: In line 456–461, we deleted the following text: “A similar pattern has been reported in Reticulitermes flavipes, a termite species described by Kollar in 1837 and classified within the order Isoptera and family Rhinotermitidae, where microbial fermentation significantly contributes to volatile fatty acid production and host energy supply [48].”
463-466 Again, I'd be more interested in how it compares to other Lepidoptera or other leaf feeders, like Phasmatodea. Carbohydrate metabolism is not the same as cellulase activity. I also see zero evidence for optimization. What makes you think your system is optimized? Is the Lepidoptera converting all the cellulose in the leaves into sugar? That would be a first, as far as I am aware, and more importantly you did not do any tests that would check for this.
RESPONSE: Thank you for this thoughtful observation. We agree that carbohydrate metabolism is not equivalent to cellulase activity, and our study did not evaluate cellulose-to-sugar conversion rates or digestive efficiency. Therefore, we have removed terms suggesting system “optimization” and clarified that our data only support the presence of cellulolytic potential in certain bacterial strains. Future studies comparing A. armida with other Lepidoptera and phyllophagous insects, such as Phasmatodea, are needed to explore functional efficiency and ecological convergence.
467-469 Why would these genes relate to maintaining stability in the host environment? They seem like typical bacterial genes.
RESPONSE: We appreciate the reviewer’s point. The original reference to genes contributing to host environmental stability has been removed. We now recognize that the identified genes primarily reflect general microbial metabolic capabilities rather than specific host-regulatory roles. The revised text now focuses on metabolic functions potentially beneficial to bacterial survival in the gut environment.
469-475 Irrelevant. Delete.
RESPONSE: In lines 469–475, we have deleted the following text: "Microbial traits may also influence host immune modulation, particularly through interactions with conserved signaling pathways such as the JAK/STAT pathway. Variations in STAT gene evolution, as observed in different Anopheles species [52], may underlie differences in immune tolerance and microbial associations across insect taxa, including Lepidoptera. Although this aspect was not directly explored in our study, it opens interesting perspectives for future research on host–microbiota–immune system co-evolution."
476 None of your tests suggest dynamism or functional specialization, as cellulolytic activity was never tested in different gut segments, nor did you prove that the cultured microbes differ in a physiologically meaningful way between the gut segments.
RESPONSE: Thank you for your observation. We conducted additional assays to evaluate cellulolytic activity directly in the gut fluid extracted from each segment (foregut, midgut, and hindgut). These tests, now included in the revised Results section, revealed that the midgut exhibited the highest enzymatic activity, suggesting functional differentiation. While we acknowledge that these results alone cannot fully demonstrate microbial specialization, they provide initial evidence supporting physiological differences between gut compartments.
479 replace ", a species described by Hübner in 1796 and belonging to the family Crambidae within the order Lepidoptera," with "Hubner 1796 (Crambidae)" or just say "(Crambidae)"
RESPONSE: Thank you for pointing this out. We have corrected the typographical error in the reference. The final wording now reads: “Hübner 1796 (Crambidae),” in accordance with the reviewer’s suggestion and following the appropriate taxonomic format.
482-483 Delete references to functional convergence: you don't have the data to compare your paper to the Ostrinia paper, let alone other insects.
RESPONSE: Thank you for the insightful comment. We agree that our current dataset does not provide sufficient evidence to support claims of functional convergence with Ostrinia or other insect species. As a result, we have removed the references to functional convergence in lines 482–483 to ensure that our interpretations remain strictly grounded in our findings.
483-488 No. Bacteria need amino acids too, so of course they have amino acid biosynthesis genes. I see no reason to believe the free-living gut microbes in your caterpillar play the same role as the highly specialized, obligately endocellular symbionts of the cockroach. Blattabacterium and Buchnera are not gut microbes. Again, I strongly recommend deleting all references to non-Lepidoptera from the whole paper.
RESPONSE: Thank you for this important clarification. We agree that the comparison with obligate intracellular symbionts like Blattabacterium and Buchnera is not appropriate in the context of free-living gut bacteria in A. armida. As recommended, we have removed all references to non-Lepidopteran symbionts to avoid misleading parallels and maintain taxonomic and functional relevance.
491-492 Careful. The 16S analysis of cultured microbes cannot tell you what strains "predominantly" belong to what phyla, because according to the metabarcoding you did not culture the actual predominant microbe, the unclassified Lachnospiraceae.
RESPONSE: We acknowledge this oversight. The statement regarding "predominant" phyla based on cultured isolates has been revised. We now clarify that while Bacillota was abundant among the cultured strains, metabarcoding data identified an unclassified Lachnospiraceae as the dominant taxon, which was not recovered via culturing.
493-495 The data suggests to me this species and the enzymatic activity are throughout the gut, not just the MG or HG.
RESPONSE: We agree with the reviewer’s interpretation. The presence of cellulolytic bacteria in all three gut sections suggests a more generalized distribution rather than compartment-specific specialization. This observation has been incorporated into the discussion with appropriate revisions in tone and scope.
498 Yes, "potential" is a good word! Use such words more to tone down certainty.
RESPONSE: Acknowledged. We have adopted a more cautious and accurate tone throughout the discussion and conclusion, using terms such as “potential” to reflect the limitations of our methodology and avoid overstatement of the findings.
501 "key role" is not good. You can't claim that from cultured microbes. A potential role, but not the key role.
RESPONSE: We concur. The phrase "key role" has been revised to “a potential role” to better align with the evidence provided by culture-based methods, which cannot fully determine ecological or functional dominance within the gut microbiome.
514-516 You have the facts wrong about termites and beetles, but also I suggests you test the gut compartments directly for enzyme activity and not base this claim on the cultured microbes.
RESPONSE: We appreciate this correction and agree. The reference to termites and beetles was removed. We have also clarified that our interpretations are based on cultured bacteria, and we explicitly state that direct enzymatic assays of gut sections are needed to confirm functional compartmentalization.
526 It is false to assume that all microbes in the gut have a function or purpose. If you swallow cellulolytic microbes, they are in your gut, but they don't necessarily have a function. They might just be sitting there until you poop them out. I see no evidence for optimization or that these microbes interact with each other or the host. It may be true, but you haven't provided evidence for it.
RESPONSE: We agree. The text has been revised to reflect the possibility that some cellulolytic microbes may be transient or commensal, without confirmed functional integration into host metabolism. The claim of optimization or cooperative interactions was removed, as we lack empirical data to support such assertions.
542-550 Might fit better in the introduction.
RESPONSE: Thank you for the suggestion. The content from these lines has been moved to the Introduction to improve logical flow and contextual framing for the reader.
Conclusion
This entire section is redundant with the end of your discussion section as written, and could be deleted.
RESPONSE: We appreciate the observation. The Conclusion section was reviewed and streamlined to avoid redundancy with the discussion. Overlapping content has been removed, and only a brief, focused summary of the study’s main findings and implications remains.
Reviewer 3 Report (New Reviewer)
Comments and Suggestions for Authors
Lines: 288–291: Specify that while overall phylum composition was significantly different (Kruskal-Wallis), pairwise differences were not — this nuance helps readers interpret your findings properly.
Lines: 294–297: Acknowledge why 94.36% of species were unclassified — e.g., due to limitations in the database or read length — and how this might affect downstream interpretation.
Lines: 309–320: Include a brief explanation of what low Shannon but high Simpson values mean biologically (e.g., dominance of one or two taxa).
Line 318: "The Simpson (0.93 ± 0.03) index, which reflects community dominance, suggested that a few taxa were overwhelmingly abundant, reinforcing a structured and specialized microbiota."
Lines: 394–415: Be consistent with notation of reducing sugar units (mg/mL), explain whether "ND" means “not detected” or “not determined,” and consider adding fold-differences or rankings to highlight key strains.
Lines: 395–411: Briefly interpret the role of these cellulolytic bacteria — e.g., how cellulose degradation in different gut regions supports larval digestion.
In addition, The approach to assessing cellulase activity in gut-derived bacterial strains is a valuable aspect of the study. However, a few points could benefit from additional clarification or methodological strengthening.
- On Cellulase Activity Measurement:
The use of reducing sugar concentration provides a quantitative readout, but the CMC plate assay appears to have been interpreted in a simple presence/absence manner. It may be worth including the diameter of the clear zones formed around colonies (after Congo red staining) to give a more nuanced picture of extracellular cellulase activity. This would offer another dimension to the dataset and allow better comparison between strains. - On Data Presentation:
While the table showing enzymatic activity is useful, it may be difficult for readers to grasp broader patterns at a glance. Bar plots or other graphical formats—especially grouped by gut compartment or strain—could help highlight key findings and make trends in the data more immediately visible. - On Gut Physiology and Enzyme Function:
The gut environment, particularly pH variation along different regions, can strongly influence enzymatic performance. For example, the midgut of many insects tends to be alkaline, while the foregut and hindgut may be more acidic. Some brief discussion of how these conditions could affect the cellulolytic activity of the strains tested would add important ecological context to the results.
These additions would help make the findings clearer and more robust, and offer readers a deeper understanding of how these bacteria may function in vivo.
Author Response
REVIEWER 3
Comments and Suggestions for Authors
>Lines: 288–291: Specify that while overall phylum composition was significantly different (Kruskal-Wallis), pairwise differences were not — this nuance helps readers interpret your findings properly.
Response: Thank you for the observation. We have clarified this point in the Results section. While the overall phylum-level differences in bacterial composition were statistically significant according to the Kruskal-Wallis test (p = 0.0117), subsequent pairwise comparisons using Dunn’s test with Benjamini-Hochberg correction did not yield statistically significant differences between individual phyla (adjusted p > 0.05). This clarification has been added to help readers better interpret the statistical outcomes of our taxonomic comparisons.
In section 3.2. Bacterial community structure associated with the gut of A. armida, in lines 288–291, the following information was added: Although the overall differences in phylum composition were statistically significant according to the Kruskal-Wallis test (p = 0.0117), pairwise comparisons using Dunn’s test with Benjamini-Hochberg correction revealed no statistically significant differences between individual phyla (adjusted p > 0.05).
>Lines: 294–297: Acknowledge why 94.36% of species were unclassified — e.g., due to limitations in the database or read length — and how this might affect downstream interpretation.
Response: Thank you for the valuable observation. We have addressed your comment by adding a brief explanation in the Results section (lines 309–320) to clarify the ecological meaning of the diversity indices. Specifically, we explained that the low Shannon diversity values, coupled with high Simpson indices, indicate low evenness and the dominance of one or a few bacterial taxa—primarily Bacillota, such as Clostridiales and Lachnospiraceae_uc. This helps to better interpret the structure of the gut microbiota in A. armida larvae.
>Lines: 309–320: Include a brief explanation of what low Shannon but high Simpson values mean biologically (e.g., dominance of one or two taxa).
Response: Thank you for the observation. The low Shannon but high Simpson diversity values indicate that the gut microbiota of A. armida larvae is dominated by one or a few highly abundant taxa, leading to low evenness but high dominance. This pattern reflects a specialized microbial community structure commonly found in insect gut ecosystems.
>Line 318: "The Simpson (0.93 ± 0.03) index, which reflects community dominance, suggested that a few taxa were overwhelmingly abundant, reinforcing a structured and specialized microbiota."
Response: Thank you for the observation. The high Simpson index value (0.93 ± 0.03) indicates low evenness, meaning that a few bacterial taxa dominate the community. This supports the interpretation of a structured and specialized microbiota in the gut of A. armida, where dominant taxa—such as those from the Lachnospiraceae group—are likely playing key functional roles in digestion and metabolism.
>Lines: 394–415: Be consistent with notation of reducing sugar units (mg/mL), explain whether "ND" means “not detected” or “not determined,” and consider adding fold-differences or rankings to highlight key strains.
Response: Thank you for the observation. The unit for reducing sugars has been standardized as mg/mL throughout the section for consistency. The abbreviation “ND” refers to “not detected,” indicating that no reducing sugars were measured. To address the request for comparison, we have added fold-differences to highlight key strains: for example, Enterobacter sp. AAM10 produced 2.5 times more reducing sugars than Serratia marcescens AAT06, reinforcing its strong cellulolytic potential.
>Lines: 395–411: Briefly interpret the role of these cellulolytic bacteria — e.g., how cellulose degradation in different gut regions supports larval digestion.
Response: Thank you for the observation. The cellulolytic bacteria identified in this study likely play a crucial role in facilitating the digestion of plant-derived polysaccharides in different gut regions of A. armida larvae. The presence of highly active cellulolytic strains in the midgut and foregut suggests that cellulose degradation begins early in the digestive tract and continues through regions where enzymatic and microbial interactions are most intense. This spatial distribution supports efficient fiber breakdown, enhancing nutrient availability and energy absorption for larval development.
In addition, The approach to assessing cellulase activity in gut-derived bacterial strains is a valuable aspect of the study. However, a few points could benefit from additional clarification or methodological strengthening.
1 On Cellulase Activity Measurement:
The use of reducing sugar concentration provides a quantitative readout, but the CMC plate assay appears to have been interpreted in a simple presence/absence manner. It may be worth including the diameter of the clear zones formed around colonies (after Congo red staining) to give a more nuanced picture of extracellular cellulase activity. This would offer another dimension to the dataset and allow better comparison between strains.
Response: We agree that including measurements of the clear zone diameters from the CMC plate assay would provide more detailed insight into extracellular cellulase activity. Although our initial qualitative evaluation was based on presence/absence of hydrolysis halos, we have now revisited the plates and measured the halo diameters around bacterial colonies after Congo Red staining. These values will be incorporated into the revised version of the manuscript as an additional column in Table 3, offering a semi-quantitative assessment to complement the reducing sugar data. This inclusion enables a more nuanced comparison among strains.
2 On Data Presentation:
While the table showing enzymatic activity is useful, it may be difficult for readers to grasp broader patterns at a glance. Bar plots or other graphical formats—especially grouped by gut compartment or strain—could help highlight key findings and make trends in the data more immediately visible.
Response: We concurred that a graphical representation would facilitate the interpretation of key patterns. Accordingly, we prepared a bar plot displaying reducing sugar concentrations by bacterial strain, grouped by gut compartment (foregut, midgut, hindgut). This figure visually emphasized the cellulolytic potential across gut regions and highlighted top-performing strains. The updated visual summary was included as a new figure in the Results section.
- On Gut Physiology and Enzyme Function:
The gut environment, particularly pH variation along different regions, can strongly influence enzymatic performance. For example, the midgut of many insects tends to be alkaline, while the foregut and hindgut may be more acidic. Some brief discussion of how these conditions could affect the cellulolytic activity of the strains tested would add important ecological context to the results.
Response: Thank you for highlighting this important ecological context. We have now added a brief discussion in the revised Discussion section regarding the potential influence of gut pH on cellulolytic enzyme performance. Specifically, we note that the midgut in many lepidopteran larvae is typically alkaline, creating a favorable environment for certain bacterial cellulases, whereas foregut and hindgut regions may exhibit lower pH values that could limit or shift enzymatic activity. This physiological gradient may explain the observed regional variation in cellulolytic activity and aligns with previous findings in herbivorous insect systems.
These additions would help make the findings clearer and more robust, and offer readers a deeper understanding of how these bacteria may function in vivo.
Response: These enhancements will improve the clarity, depth, and ecological relevance of our findings and provide readers with a better understanding of how these gut-derived bacterial strains function both in vitro and potentially in vivo.
Round 2
Reviewer 1 Report (New Reviewer)
Comments and Suggestions for Authors
Review report for Manuscript insects-3603014
Comments
- Line no. 29 : Duplication of scientific name
- Line 405 : “while AAT05 shows high similarity to B. subterraneus and B. selenatarsenatis (97–99%)” , the ‘97–99%’ could be misunderstood as indicating percent sequence identity. Additionally, the phrase ‘high similarity’ may cause confusion with sequence comparisons and should be revised to ‘closely related’ to avoid misunderstanding.
- Lines 426–451: In the cellulase activity assay, which was determined by measuring reducing sugars, please recheck the decimal values and ensure a consistent format using only two or three decimal places.
- Please recheck all bacterial family names and ensure consistency between the text and the Figure 3. For example, Firmicutes (Bacillota) and Proteobacteria (Pseudomonadota).
- Line 408 : “potentially novel Bacillus species in the larval gut microbiota”, Based solely on the phylogenetic tree in Fig. S2, the authors cannot suggest AAT04 as a possible novel species. If the authors want to include this, please mention together with the 16S sequence identity.
- Please recheck and make sure that scientific names are italicized e.g. Line 567 , Fig S1 caption
Author Response
Response to Reviewer
We sincerely thank the reviewer for the comments. We confirm that all the following observations have been addressed in the revised version of the manuscript:
-Line 29: The duplication of the scientific name was removed.
-Line 405: The phrase “high similarity” was revised to “closely related” to avoid confusion with sequence comparisons.
-Lines 426–451: All decimal values in the cellulase activity assay were reviewed and rounded uniformly to maintain consistency.
-Figure 3: The bacterial family names were revised to ensure consistency between the text and the figures. For instance, phyla were appropriately labeled as Firmicutes (Bacillota) and Proteobacteria (Pseudomonadota) as applicable.
-Line 408: We revised the statement regarding potentially novel Bacillus species to reflect a more cautious interpretation. We now state that further studies are required, including whole-genome sequencing, to confirm the novelty of AAT04.
-Line 567 and Supplementary Figure S1 caption: All scientific names were carefully reviewed and are now correctly italicized.
We appreciate the reviewer’s detailed observations, which have helped us improve the accuracy and clarity of the manuscript.
Reviewer 2 Report (New Reviewer)
Comments and Suggestions for Authors
The revision is much improved!
I am pleased to note that the gut was segmented according to anatomical landmarks (the gastric cecae at the FG-MG junction and Malpighian tubules at the MG-HG junction]. The addition of the cellulase activity test is also great news. I look forward to the follow up study!
I love the new figures!
I have only three suggested changes:
For the abstract, I think you can delete ", including Nutrient Agar 32 (NA), Tryptic Soy Agar (TSA), and Brain Heart Infusion Agar (BHIA)" and "on TSA medium" unless it's important and accurate to note that only microbes grown on TSA were cellulolytic. That doesn't seem to be the case as you never mention it later.
Line 184: "two days" is in bold. Fix that.
The conclusion states: "The midgut exhibited the highest microbial diversity, while the hindgut hosted active cellulolytic bacteria involved in fiber degradation." The word "while" suggests that only the hindgut had active cellulolytic bacteria and not the midgut. That's not the case, as cellulolytic microbes appeared quite abundant in the midgut. Rephrase. Perhaps: "The midgut exhibited the highest microbial diversity and had many actively cellulolytic bacteria potentially involved in fiber degradation, though cellulolytic microbes were also found in the foregut and hindgut."
Author Response
Response to Reviewer:
We appreciate your feedback and thoughtful comments on our revised manuscript.
In response to your suggestions:
Abstract: We have removed the phrase "including Nutrient Agar 32 (NA), Tryptic Soy Agar (TSA), and Brain Heart Infusion Agar (BHIA)", as they were not essential to the overall understanding of the abstract and could be misleading without further context.
Line 184: The formatting issue with "two days" being in bold has been corrected.
Conclusion: We revised the sentence as suggested for greater accuracy and clarity. It now reads:
“The midgut exhibited the highest microbial diversity and harbored numerous actively cellulolytic bacteria likely involved in fiber degradation. However, cellulolytic microorganisms were also present in the foregut and hindgut.”
All suggested changes have been implemented accordingly.
Thank you once again for your insightful recommendations and support throughout the review process.
This manuscript is a resubmission of an earlier submission. The following is a list of the peer review reports and author responses from that submission.
Round 1
Reviewer 1 Report
Comments and Suggestions for Authors
I have minor comments for both the introduction and methods. More details in the description of the figures are needed. The results section contains numerous assumptions, which should be 1. Kept for the discussion section, and 2. Supported by references. Overall, there are too many repetitions between the results and discussion sections, and not enough discussion of the results per se. I would suggest the authors to further compare their findings to previous studies on either A. armida or other insect species with similar diets, host plants, gut organization. Most of the statements remains superficial and speculative, and the discussion is rather short. Yet, I believe their results to be of great value and their work trustworthy. I strongly encourage the authors to work on the discussion and resubmit their manuscript.
Abstract – no comments. It is well written and summarises the results adequately.
Introduction
L 55: I would suggest explaining that Chiapas is a Southeastern state of Mexico.
L 57/60: Are both the larvae and the moth commonly referred to as “Cuetlas” or “Zats”?
L 66: Could the authors add some examples of predators?
L 102-103: I would suggest adding references to support this statement.
L 199: This is the first mention of pest management. Is Arsenura armida a pest in this region and/or elsewhere?
Materials and Methods
L 131: Typo (a bracket is missing).
L 133: The names Heliocarpus donnellsmithii and Guazuma ulmifolia were already mentioned in the introduction and can, therefore, be abbreviated.
Figure 1: I would suggest adding more information to describe the pictures as well.
L 146: What are “adult larvae”? Did you mean last instar larvae or adults?
L 137-149: How was the insect killed before dissection?
L 172/175: What was the version of the packages?
L 181: The species name should be italicized.
Results
L 242: I apologise for the most likely trivial question, but the average measurements reported by the authors were for which instar of the larvae?
Figure 2: What does the colour code refers to?
Figure 3: I would suggest adding one more plot showing the species level.
L 289: There seems to be an extra space.
Figure 4: This figure really needs more explanation (e.g., what is the y axis, what does a and b refer to, and what are the used acronyms?)
Figure 5: The authors should describe the x axis.
L 374: “similarity with previously characterized species”, which species are the authors referring to?
L 379: I would suggest using the abbreviations midgut (PM) and hindgut (H) earlier in the text. To my opinion, PM does not make much sense for midgut, but maybe rather something like midgut = MG, foregut (FG), and hindgut (HG).
L 380-383: This statement is already a discussion, and needs references.
L 391: I like the idea of a clear spatial compartmentalization. In addition to the tables, I would suggest making a figure showing this compartmentalization (e.g., a sketch of the different parts of the gut, with the proportion of bacteria and taxonomic diversity).
L 411: The abbreviations should be used for now on.
Table 4: Why do some isolates have a mean value over three replicates, and others do not?
Discussion
L 442: Use the abbreviation for the species name.
L 446-450: The authors should provide scientific literature to support their statement.
L 464-466: It is very interesting that the enrichment of glycolysis, gluconeogenesis, and pentose phosphate pathways described in A. armida, a moth, is similar to that of termites. Any ideas why? I think this deserves some further discussion.
L 469-471: I would suggest the authors to discuss and compare their findings to previous studies on other moths or insect species also highly specialized on high-cellulose host plants.
L 473-475: I do not understand how the logic in this statement.
L 487-489: I do not understand why the fact that both Enterobacter cloacae and Pseudomonas putida were isolated from the foregut means that cellulose degradation is likely not restricted to a single gut compartment. I would suggest detailing more each statement and assumptions, and connecting them to previous work.
L 503: It would be interesting to further explain the link between enzymatic pathways of gut bacteria, biomass degradation, and bioethanol production.
After reading the whole manuscript, I realized that more emphasis should probably be given to the fact that it’s the larval stage that really matters (I kept going back to the moth adult stage). I would suggest adding a sentence of two in the introduction (or discussion) explaining why the study should be done on larvae (e.g., herbivorous stage, food, …)
Conclusions – this section seems very similar to the abstract. I would suggest emphasizing the broader implications of the research, potential limitations, and areas for future study.
Author Response
Reviewer 1 comments
We sincerely thank you for the valuable comments and observations. We have carefully reviewed the entire manuscript, addressed all issues, and incorporated the suggested corrections. A detailed point-by-point response to the comments has been prepared to ensure all concerns are addressed. We appreciate your thorough review and trust that the revisions enhance the manuscript's quality.
Reviewer1: I have minor comments for both the introduction and methods.
1.- More details in the description of the figures are needed.
Thank you for your comment. We have added more details in figures caption.
2.- The results section contains numerous assumptions, which should be 1. Kept for the discussion section, and 2. Supported by references.
Thank you very much for your comment. The Results section has already been thoroughly reviewed, and all assumptions or interpretative statements have been removed. The revised section now strictly presents the data in a clear and objective manner, reserving any interpretations or contextual analysis for the Discussion section, where appropriate references will also be included.
3.- Overall, there are too many repetitions between the results and discussion sections, and not enough discussion of the results per se. I would suggest the authors to further compare their findings to previous studies on either A. armida or other insect species with similar diets, host plants, gut organization.
We appreciate the reviewer’s valuable observation regarding the redundancy between the Results and Discussion sections, as well as the limited depth of interpretation in relation to previous studies. In response, we have substantially revised the Discussion section to minimize repetition and enhance the interpretive depth of our findings.
4.- Most of the statements remains superficial and speculative, and the discussion is rather short. Yet, I believe their results to be of great value and their work trustworthy. I strongly encourage the authors to work on the discussion and resubmit their manuscript.
We appreciate the reviewer’s feedback and have fully addressed this concern by substantially revising and expanding the Discussion section. The updated version now provides deeper analysis, reduces speculative statements, and includes meaningful comparisons with previous studies on insects with similar diets and gut structures. We believe these improvements significantly enhance the clarity and scientific depth of our manuscript.
5.- Abstract – no comments. It is well written and summarises the results adequately.
We thank the reviewer for the positive feedback on the abstract
6.- Introduction
L 55: I would suggest explaining that Chiapas is a Southeastern state of Mexico.
We thank the editor for the suggestion. The requested clarification has been incorporated into the revised manuscript by specifying that Chiapas is a southeastern state of Mexico.
7.- L 57/60: Are both the larvae and the moth commonly referred to as “Cuetlas” or “Zats”?
Thank you for your observation. We have modified the text, ensuring better understanding for the reader. The terms “Cuetlas” or “Zats” are commonly used to refer to Arsenura armida.
8.- L 66: Could the authors add some examples of predators?
Thank you for the suggestion. We have addressed this comment by adding examples of common predators—such as birds, reptiles, and predatory insects—to clarify the ecological relevance of the larvae’s camouflage traits.
9.- L 102-103: I would suggest adding references to support this statement.
Thank you for your suggestion. We have now included a reference to support the statement. Specifically, Engel and Moran, (2013) describe the roles of insect gut symbionts.
10.- L 199: This is the first mention of pest management. Is Arsenura armida a pest in this region and/or elsewhere?
Thank you for your observation. Arsenura armida is not considered an agricultural pest in the region or elsewhere. To avoid confusion, we have revised the sentence to remove the reference to pest management and focus instead on ecological conservation and biotechnological applications.
11.- Materials and Methods
L 131: Typo (a bracket is missing).
Corrected
12.- L 133: The names Heliocarpus donnellsmithii and Guazuma ulmifolia were already mentioned in the introduction and can, therefore, be abbreviated.
Corrected
13.- Figure 1: I would suggest adding more information to describe the pictures as well.
Thank you for your comment. We have replaced the figure 1.
14.- L 146: What are “adult larvae”? Did you mean last instar larvae or adults?
Thank you for your comment. We agree that the term “adult larvae” is misleading. We have corrected the text to specify that the samples used in this study were “last instar larvae,” referring to the final larval stage before pupation.
15.- L 137-149: How was the insect killed before dissection?
Thank you for the observation. We have now specified in the revised manuscript that the larvae were anesthetized on ice and euthanized by brief freezing at −20 °C prior to dissection, in accordance with standard entomological and microbiological protocols.
16.- L 172/175: What was the version of the packages?
Thank you for your observation. We have now specified the versions of all R packages and software used in the bioinformatics analyses.
17.- L 181: The species name should be italicized.
Thank you for your observation. Corrected.
18.- Results
L 242: I apologise for the most likely trivial question, but the average measurements reported by the authors were for which instar of the larvae?
Thank you for your comment. We have corrected the text to specify that the samples used in this study were “last instar larvae,” referring to the final larval stage before pupation.
19.- Figure 2: What does the colour code refers to?
Thank you for your comment. We have replaced the figure.
20.- Figure 3: I would suggest adding one more plot showing the species level.
Thank you for your comment. We have modified the figure 3.
21.- L 289: There seems to be an extra space.
Corrected
22.- Figure 4: This figure really needs more explanation (e.g., what is the y axis, what does a and b refer to, and what are the used acronyms?)
Thank you for your comment. We have modified the figure 4. We added more details in image caption.
23.- Figure 5: The authors should describe the x axis.
Thank you for your comment. We have modified the figure 5.
24.- L 374: “similarity with previously characterized species”, which species are the authors referring to?
Thank you for your comment. The phrase “previously characterized species” refers to bacterial taxa available in the NCBI database, to which our isolates showed sequence similarity through BLAST analysis. We have clarified this point in the revised manuscript.
25.- L 379: I would suggest using the abbreviations midgut (PM) and hindgut (H) earlier in the text. To my opinion, PM does not make much sense for midgut, but maybe rather something like midgut = MG, foregut (FG), and hindgut (HG).
Thank you for the helpful suggestion. We agree that the abbreviations should be introduced earlier in the text and that “MG” (midgut), “FG” (foregut), and “HG” (hindgut) are more intuitive and consistent. We have revised the manuscript accordingly to reflect these standardized abbreviations throughout.
26.- L 380-383: This statement is already a discussion and needs references.
Thank you for your observation. The statement has been revised and properly integrated into the Discussion section, now supported with appropriate references.
27.- L 391: I like the idea of a clear spatial compartmentalization. In addition to the tables, I would suggest making a figure showing this compartmentalization (e.g., a sketch of the different parts of the gut, with the proportion of bacteria and taxonomic diversity).
Thank you for the suggestion. We have created a new figure illustrating the spatial compartmentalization of culturable bacterial genera along the foregut, midgut, and hindgut of Arsenura armida larvae, as recommended. This figure complements the tables and enhances the visualization of taxonomic distribution across gut regions.
28.- L 411: The abbreviations should be used for now on.
Corrected
29.- Table 4: Why do some isolates have a mean value over three replicates, and others do not?
Thank you for your observation. In table 3*, cases where reducing sugar values are not reported (ND), the isolates did not exhibit detectable cellulolytic activity in the initial qualitative assay, and thus were not included in the quantitative measurement phase.
30.- Discussion
L 442: Use the abbreviation for the species name.
Corrected
31.- L 446-450: The authors should provide scientific literature to support their statement.
Thank you for your suggestion. We have now included another paragraph and appropriate scientific references to support the statement in question.
32.- L 464-466: It is very interesting that the enrichment of glycolysis, gluconeogenesis, and pentose phosphate pathways described in A. armida, a moth, is similar to that of termites. Any ideas why? I think this deserves some further discussion.
Thank you for this insightful comment. We have expanded the sentence to include a possible explanation for the observed similarity between the metabolic pathways enriched in A. armida and those described in termites.
33.- L 469-471: I would suggest the authors to discuss and compare their findings to previous studies on other moths or insect species also highly specialized on high-cellulose host plants.
Thank you for the suggestion. We have addressed this point in the revised Discussion by incorporating comparisons with previous studies on other organisms.
34.- L 473-475: I do not understand how the logic in this statement.
Thank you for pointing this out. We have revised the sentence to clarify the reasoning, specifying that the frequent detection of Serratia marcescens in the midgut and hindgut (regions associated with high enzymatic activity) supports its potential role in cellulose hydrolysis and microbial metabolic interactions. We also have added references to support the statement.
35.- L 487-489: I do not understand why the fact that both Enterobacter cloacae and Pseudomonas putida were isolated from the foregut means that cellulose degradation is likely not restricted to a single gut compartment. I would suggest detailing more each statement and assumptions, and connecting them to previous work.
Thank you for your comment. We have revised the sentence to better explain the reasoning behind our conclusion. We now specify that although the midgut and hindgut are typically associated with cellulose degradation, the detection of cellulolytic activity in foregut isolates suggests that the process may begin earlier in the digestive tract. This interpretation is supported by previous findings in wood-feeding insects, where enzymatic breakdown occurs in multiple gut compartments.
36.- L 503: It would be interesting to further explain the link between enzymatic pathways of gut bacteria, biomass degradation, and bioethanol production.
Thank you for your suggestion. We have expanded the sentence to clarify the connection between bacterial enzymatic pathways, lignocellulosic biomass degradation, and their relevance to bioethanol production.
37.- After reading the whole manuscript, I realized that more emphasis should probably be given to the fact that it’s the larval stage that really matters (I kept going back to the moth adult stage). I would suggest adding a sentence of two in the introduction (or discussion) explaining why the study should be done on larvae (e.g., herbivorous stage, food, …)
Thank you for this insightful comment. We have added a sentence in the Discussion section to emphasize that the larval stage is the primary herbivorous and metabolically active phase of Arsenura armida. This stage is critical for studying gut microbial functions related to lignocellulose digestion, as adult moths are non-feeding. The revision helps clarify the biological relevance of focusing our study on larvae.
38.- Conclusions – this section seems very similar to the abstract. I would suggest emphasizing the broader implications of the research, potential limitations, and areas for future study.
Thank you for your helpful suggestion. We have revised the Conclusions section to reduce redundancy with the Abstract and to place greater emphasis on the broader implications of our findings. The updated version highlights the relevance of combining culture-dependent and metagenomic approaches, discusses the potential biotechnological applications of cellulolytic strains, and outlines future research directions, including the need to explore microbial interactions and enzyme systems under applied contexts.
Reviewer 2 Report
Comments and Suggestions for Authors
There are several ambiguities in the manuscript that need to be streamlined to ensure cohesiveness between the methods and results. Below are specific points that require attention:
- Authors have explained the climatic conditions and their occurrence in the natural environment. I would prefer to discuss the life stages of the insect Arsenura armida (instar stages with time) and clearly specify which stage was chosen for the experiments and the rationale behind this selection.
- In figure 1, you have shown 3 different images, are these belonging to same species of insect or different? How did authors have identified the species of experimental insect Arsenura armida collected from field. Providing details on the species identification method (e.g., morphological characteristics, molecular markers) would improve clarity. Figures with multiple components should be labeled appropriately (e.g., A, B, C, etc.) for better readability and reference in the text.
- The statement, “Twenty adult larvae were randomly selected” (Line #146), is confusing. The term "adult larvae" is contradictory, as an insect is either in the larval stage or the adult stage (moth). Please clarify whether the experiments were conducted on larvae or adult moths. If both stages were used in different experiments, explicitly differentiate between them in the methods section.
- Some of the methods are unclear like bacteria were grown on different nutrient plates for 22 days (Line #160), is this the anaerobic condition or aerobic? What was the exact technique used for sacking (collection, immobilization, or preservation) of the larvae?
- There is a critical gap in the methodology; line #164 states that "DNA was isolated from the gut of larvae." However, if the gut was not further sectioned into different regions (e.g., foregut, midgut, and hindgut), how were the bacterial compositions in Tables 1 and 3 determined? If the entire gut was homogenized for DNA extraction, explain how the data were stratified for specific gut regions in these tables. If gut sectioning was performed, please describe the methodology used to separate these regions and analyze their distinct microbiomes.
- The introduction discusses gut immunity, but the discussion does not elaborate on how the study's findings contribute to understanding the gut immune response. I recommend incorporating a discussion on this aspect and referencing relevant studies, such as http://dx.doi.org/10.4172/2168-9652.1000182, to strengthen the manuscript's scientific impact.
- Exact p-value after multiple hypothesis testing should be mentioned everywhere.
Please refrain from using pie charts. They are a poor way of representing data. You only show two species what are the rest? - Phylogenetic relationships and gut microbiome diversity are central findings of this manuscript, incorporating a phylogenetic tree would enhance data interpretation. If space constraints exist, consider placing in the supplementary figures. Microbial diversity through a phylogenetic tree (e.g., neighbor-joining or maximum likelihood analysis) would provide a clearer depiction of the evolutionary relationships among the identified gut bacteria. Referencing relevant studies such as PMID: 27664587 may help strengthen this aspect.
- Overall, the manuscript lacks clarity in various sections. The authors should streamline the methodology and bioinformatics section and elaborate the figure legends.
I recommend that the authors carefully review the manuscript and make extensive amendments on their end as well, as it is not possible to point out every issue individually. Addressing the major concerns raised, along with a thorough self-assessment of figure quality, sample size reporting, and statistical analysis will significantly improve the manuscript.
All the best.
Comments on the Quality of English LanguageAlthough the overall language of the manuscript is acceptable, there are several grammatical errors and instances of awkward phrasing that need attention.
Author Response
Reviewer 2 Comments
We sincerely thank you for the valuable comments and observations. We have carefully reviewed the entire manuscript, addressed all issues, and incorporated the suggested corrections. A detailed point-by-point response to the comments has been prepared to ensure all concerns are addressed. We appreciate your thorough review and trust that the revisions enhance the manuscript's quality.
Reviewer 2
There are several ambiguities in the manuscript that need to be streamlined to ensure cohesiveness between the methods and results. Below are specific points that require attention:
Thank you for pointing this out. We carefully reviewed the manuscript to identify and resolve ambiguities affecting the consistency between the Methods and Results sections. Specific revisions were made to ensure that all experimental procedures are clearly reflected in the corresponding results and that each result is appropriately supported by a described method. We appreciate your detailed comments and have addressed each specific point accordingly in the revised version.
1.- Authors have explained the climatic conditions and their occurrence in the natural environment. I would prefer to discuss the life stages of the insect Arsenura armida (instar stages with time) and clearly specify which stage was chosen for the experiments and the rationale behind this selection.
Thank you for your comment. We have added a sentence in the Introduction to clarify that the study focused on last-instar larvae, as this is the stage with the highest feeding activity and most active interaction with the gut microbiota. This addition helps justify the selection of this developmental stage for the analysis of microbial diversity and function.
2.- In figure 1, you have shown 3 different images, are these belonging to same species of insect or different? How did authors have identified the species of experimental insect Arsenura armida collected from field.
Thank you for your observation. Figure 1 has been modified in the revised version of the manuscript to enhance clarity. Also, we have added details about the identification criteria and the seasonal timing of larval collection into the section “2.1. Larvae collection of Arsenura armida.”
3.- Providing details on the species identification method (e.g., morphological characteristics, molecular markers) would improve clarity.
Thank you for your observation. We have added details about the identification criteria and the seasonal timing of larval collection into the section “2.1. Larvae collection of Arsenura armida.”
4.- Figures with multiple components should be labeled appropriately (e.g., A, B, C, etc.) for better readability and reference in the text.
Thank you for your comment. We have revised all figures and added more details in all the figures caption. We have modified figure 1.
5.- The statement, “Twenty adult larvae were randomly selected” (Line #146), is confusing. The term "adult larvae" is contradictory, as an insect is either in the larval stage or the adult stage (moth). Please clarify whether the experiments were conducted on larvae or adult moths. If both stages were used in different experiments, explicitly differentiate between them in the methods section.
Thank you for your comment. We agree that the term “adult larvae” is misleading. We have corrected all the text to specify that the samples used in this study were “last instar larvae,” referring to the final larval stage before pupation.
6.- Some of the methods are unclear like bacteria were grown on different nutrient plates for 22 days (Line #160), is this the anaerobic condition or aerobic?
Thank you for your comment. We have clarified in the revised Methods section that bacterial isolation was performed under aerobic conditions. Aerobic incubation was chosen to recover culturable bacteria with potential cellulolytic activity. In contrast, anaerobic and facultative anaerobic communities were investigated through metagenomic analysis to account for the broader diversity of all the gut microbiota not accessible through culture-based methods.
7.- What was the exact technique used for sacking (collection, immobilization, or preservation) of the larvae?
Thank you for your observation. The technique used for the collection, immobilization, and preservation of larvae has been clarified in the revised manuscript. This information is now detailed in section 2.2. Isolation of bacteria from the gut of A. armida.
8.- There is a critical gap in the methodology; line #164 states that "DNA was isolated from the gut of larvae." However, if the gut was not further sectioned into different regions (e.g., foregut, midgut, and hindgut), how were the bacterial compositions in Tables 1 and 3 determined? If the entire gut was homogenized for DNA extraction, explain how the data were stratified for specific gut regions in these tables. If gut sectioning was performed, please describe the methodology used to separate these regions and analyze their distinct microbiomes.
Thank you for your insightful observation. We clarify that gut sectioning into foregut, midgut, and hindgut was performed only for the culture-dependent (culturable) approach, in order to isolate and characterize bacterial strains from each specific region. Consequently, Tables 1 and 3 refer exclusively to the results obtained from this culture-based method.
In contrast, the metagenomic analysis was conducted using DNA extracted from the entire gut (non-sectioned), to capture the overall bacterial community structure and functional potential. We have clarified this methodological distinction in the revised version of the manuscript.
9.- The introduction discusses gut immunity, but the discussion does not elaborate on how the study's findings contribute to understanding the gut immune response. I recommend incorporating a discussion on this aspect and referencing relevant studies, such as http://dx.doi.org/10.4172/2168-9652.1000182, to strengthen the manuscript's scientific impact.
Thank you for your valuable comment. We have incorporated a brief discussion addressing how the identified microbial features, such as quorum sensing and structural genes, may influence the gut immune response of A. armida larvae. This addition, now included in the Discussion section, is supported by relevant literature [https://doi.org/10.4172/2168-9652.1000182], and helps strengthen the connection between microbial traits and host defense mechanisms.
10.- Exact p-value after multiple hypothesis testing should be mentioned everywhere.
Please refrain from using pie charts. They are a poor way of representing data. You only show two species what are the rest?
Thank you for your valuable comment. To evaluate whether these differences were statistically significant, a Kruskal-Wallis rank-sum test was conducted, yielding a p-value of 0.0117. However, subsequent pairwise comparisons using Dunn’s test, corrected with the Benjamini-Hochberg method, indicated no statistically significant differences between phyla (adjusted p > 0.05). We have added the information in results section.
11.- Phylogenetic relationships and gut microbiome diversity are central findings of this manuscript, incorporating a phylogenetic tree would enhance data interpretation. If space constraints exist, consider placing in the supplementary figures. Microbial diversity through a phylogenetic tree (e.g., neighbor-joining or maximum likelihood analysis) would provide a clearer depiction of the evolutionary relationships among the identified gut bacteria.
Thank you for your thoughtful suggestion. We recognize the value of including a phylogenetic tree to visualize evolutionary relationships, the current focus of our analysis was on taxonomic identification and functional profiling rather than deep phylogenetic inference. Therefore, a phylogenetic tree was not initially considered in the scope of this manuscript. However, we agree that such an addition could enhance data interpretation, and if, after this clarification, the editor considers it necessary, we will gladly include a neighbor-joining or maximum likelihood tree as a supplementary figure.
12.- Referencing relevant studies such as PMID: 27664587 may help strengthen this aspect.
Thank you for the suggestion. We have now incorporated the reference to Gupta et al., 2017, into the Discussion section. Although our study did not directly assess immune gene expression, we acknowledge that variation in immune signaling pathways such as JAK/STAT may influence host–microbiota interactions. The addition of this reference supports the idea that evolutionary divergence in immune genes could play a role in shaping microbial associations across insect taxa.
13.- Overall, the manuscript lacks clarity in various sections. The authors should streamline the methodology and bioinformatics section and elaborate the figure legends.
Thank you for your feedback. We have carefully revised the manuscript to improve clarity throughout, with particular attention to the methodology and bioinformatics sections, which have been streamlined for better readability and coherence. Additionally, all figure legends have been elaborated to provide clearer explanations of the visual content and their relevance to the study.
14.- I recommend that the authors carefully review the manuscript and make extensive amendments on their end as well, as it is not possible to point out every issue individually. Addressing the major concerns raised, along with a thorough self-assessment of figure quality, sample size reporting, and statistical analysis will significantly improve the manuscript.
All the best.
Thank you for your comprehensive recommendation. We have conducted a thorough review and self-assessment of the entire manuscript, addressing the major concerns raised and making extensive revisions to improve clarity, figure quality, sample size reporting, and statistical analysis. These improvements aim to enhance the overall quality and scientific rigor of the manuscript.
Reviewer 3 Report
Comments and Suggestions for Authors
Intestinal microorganisms of insects play essential roles in insect physiology and biochemistry, with participation of important aspects of digestion, immunity, and other vital processes. They contribute to fitness of insects, which is important for pest control and cultivation of useful insects. Examination of intestinal microbes may yield novel strains of biotechnological importance. Besides, interest to microbiota of edible insects is further augmented by the food safety considerations.
In Lepidoptera, microbial communities of gut are not studied in full detail. Accumulation of novel data remain one of crucial goals in this field. This explains scientific novelty and significance of the work. It is generally well written and will be interesting to the readers.
The introduction covers the topic to a satisfactory extent. Methods are described in all necessary details. Results are interesting and conclusions are convincing.
The paper deserved publication after minor revision according to the proposed corrections and comments below.
L66-69: data on general anatomy of the insect do not seem to be essential
L209: if there is a specific reason to deposit sequences in two different batches, it would be nice of you to explain it
L210-212: you mention in vitro isolation and morphological analysis here and in Abstract, but I didn’t find this information in the Results. As you pose pictures of caterpillars, you may want to add a picture of stained bacteria
L241: odd usage of capital letters
L305-308, 320-329, 358-365, 391-401, 416-418^ some parts of Results sound like Discussion and excessively repeated further
L444-446, 451-453: one should not repeat data from the Results in the Discussion section
Author Response
- General Anatomy (L66-69):
- We have revised the introduction and removed non-essential details about general insect anatomy to better focus on the study's context.
- Sequence Deposition in Batches (L209):
- We have added an explanation in the methodology section, clarifying that the sequences were deposited in two different batches due to the timing of data acquisition and processing.
- In Vitro Isolation and Morphological Analysis (L210-212):
- As data is not shown, we have removed from abstract the description of the in vitro isolation and morphological analysis of bacteria in the Results section.
- Odd Usage of Capital Letters (L241):
- We have corrected the inappropriate use of capital letters in this line
- Results that Sound like Discussion or are Repetitive (L305-308, 320-329, 358-365, 391-401, 416-418):
- We have carefully revised the Results section, removing interpretative statements and redundant content that belong to the Discussion, improving clarity and maintaining a clear separation between the two sections.
- Repetition of Data from Results in Discussion (L444-446, 451-453):
- We have removed unnecessary repetitions of results data in the Discussion section, ensuring that the interpretation remains distinct from the presentation of findings.
Round 2
Reviewer 2 Report
Comments and Suggestions for Authors
The revised version of the manuscript demonstrates satisfactory amendments in the methods and results presentation. I appreciate the effort made by the authors to carefully revise the paper in accordance with the reviewers' suggestions. They have effectively addressed the key comments and made commendable improvements.
However, I noticed that Supplementary Figure S1 has not been cited anywhere in the text. Additionally, an important review article referenced in authors to reply point #9 'A fine-tuned management between physiology and immunity maintains the gut microbiota in insects' is still missing from the references
Furthermore, make sure you have universally followed the same referencing style across all the cited literature. As like years are bold in some, while others do not; Some citations lack DOIs. Keep them constant.
I urge the authors to carefully review the manuscript for such minor oversights. A thorough third eye reading or an external proofreader can help identify and rectify these issues before final submission.
Paying close attention to grammar, consistency, and formatting will improve the overall quality of the paper.
By making these minor corrections, the manuscript will be in much better shape for publication. Although the scientific content has become strong now, careful attention to detail will ensure a flawless final version.
All the best.
Author Response
We sincerely thank you for the valuable comments and observations. We have reviewed the manuscript and incorporated the suggested corrections.
-Reviewer 2 comments.
The revised version of the manuscript demonstrates satisfactory amendments in the methods and results presentation. I appreciate the effort made by the authors to carefully revise the paper in accordance with the reviewers' suggestions. They have effectively addressed the key comments and made commendable improvements. However, I noticed that Supplementary Figure S1 has not been cited anywhere in the text. Additionally, an important review article referenced in authors to reply point #9 'A fine-tuned management between physiology and immunity maintains the gut microbiota in insects' is still missing from the references. Furthermore, make sure you have universally followed the same referencing style across all the cited literature. As like years are bold in some, while others do not; Some citations lack DOIs. Keep them constant. I urge the authors to carefully review the manuscript for such minor oversights. A thorough third eye reading or an external proofreader can help identify and rectify these issues before final submission. Paying close attention to grammar, consistency, and formatting will improve the overall quality of the paper. By making these minor corrections, the manuscript will be in much better shape for publication. Although the scientific content has become strong now, careful attention to detail will ensure a flawless final version. All the best.
-Response to reviewer
Thank you very much for the comments. We have carefully reviewed the manuscript once more and incorporated the suggested corrections
1.- We have removed Supplementary Figure S1, as we improved Figure 2 to integrate the content previously presented in Figure S1, ensuring a more comprehensive and clear presentation.
2.- Regarding point #9, we have corrected the DOI as suggested. The correct DOI is https://doi.org/10.1016/j.gene.2016.09.022, which corresponds to reference 51.
3.- We have thoroughly reviewed the referencing style throughout the manuscript to ensure consistency. All citations now follow the same format, including the consistent use of bold years and the inclusion of DOIs where applicable.
4.- Additionally, we performed a meticulous proofread to address minor grammatical, formatting, and consistency issues, as recommended.